# GeoMind: A Geometric Neural Network of State Space Model for Understanding Brain Dynamics on Riemannian Manifold

## Abstract

State space model (SSM) is a powerful tool in neuroscience field to characterize the dynamic nature of brain functions by elucidating the mechanism of how brain system transits between brain states and how underlying states give rise to the observed neural activities. Although tremendous efforts have been made to lend the power of deep learning and mathematical insight of SSM in various functional neuroimaging studies, current state-of-the-art methods lack a holistic view of brain state evolution as a self-organized dynamical system where each part of the brain is functionally inter-connected. Since the topological co-activation of functional fluctuations exhibits an intrinsic geometric pattern (symmetric and positive definite, or SPD) on the Riemannian manifold, the call for understanding how a selective set of functional connectivities in the brain supports diverse behavior and cognition emerges a new machine learning scenario of manifold-based SSM for large-scale functional neuroimages. To that end, we propose a geometric neural networks, coined *GeoMind*, designed to uncover evolving brain states by tracking the trajectory of functional dynamics on a high-dimensional Riemannian manifold of SPD matrices. Our *GeoMind* demonstrates promising results in identifying specific brain states based on task-based functional Magnetic Resonance Imaging (fMRI) data, as well as in diseases early diagnosis for Alzheimer's disease, Parkinson's disease and Autism. These results highlight the applicability of the proposed *GeoMind* in neuroscience research. Furthermore, to assess the generalization capabilities of our model, we applied it to the domain of human action recognition (HAR), achieving promising performance on three benchmark datasets (UTKinect, Florence and HDM05). This demonstrates the scalability and robustness of the proposed geometry deep model of SSM in capturing complex spatio-temporal dynamics across diverse fields.

## 1 Introduction

The human brain is a complex and dynamic system composed of distinct structural regions, each specialized for specific functions (Bassett et al., 2011; Hutchison et al., 2013). While these regions are locally segregated, they are dynamically inter-connected to process a wide range of information. Over the past few decades, understanding the functional mechanisms of human brain has been a central focus in both basic and clinical neuroscience. Functional magnetic resonance imaging (fMRI) is a popular non-invasive technique in neuroimaging field, which measures changes in blood oxygen level-dependent (BOLD) signals over time. Although converging evidence supports the biological mechanism that BOLD signals underline the neural activities, research focus has been shifted to investigate the functional connectivity (FC) which characterizes the co-activations of functional fluctuations throughout the entire brain (Bassett & Sporns, 2017).

In the majority of current functional brain network studies, Pearson's correlation is used to measure the strength of FC between two brain regions (Van Den Heuvel & Pol, 2010; Amaral et al., 2008). Recently, there has been a growing consensus in the neuroimaging field, that the topology of functional brain networks changes over time, even in a task-free environment (Bassett et al., 2011). For instance, abnormal dynamics in functional connectivity have been linked to various brain disorders, providing critical insights into the underlying neurobiological processes (Breakspear, 2017). In light of this, striking efforts have been made to uncover the neurobiological mechanism of brain activi-

ties by modeling the transit of latent brain states from the observed BOLD signals or evolving FCs (Logothetis et al., 2001; Fox et al., 2006; Pievani et al., 2014).

Functional dynamics are modeled through two main approaches: (1) leveraging temporal heuristics in BOLD signals and (2) capturing topology changes in evolving FC matrices. BOLD signals, which track blood oxygen changes, provide neural activity information but struggle to disentangle intrinsic fluctuations from external noise. For example, neural mass models in (Singh et al., 2020) describe brain dynamics using non-linear equations but often overlook spatial dependencies. In contrast, FC matrices reveal functional relationships between brain regions by correlating BOLD signals, offering insights into network-level interactions. Dynamic FC (dFC) extends this by tracking temporal connectivity evolution via sliding windows (Karahanoğlu & Van De Ville, 2017), integrating spatial and temporal information. For example, (Dan et al., 2022a) proposed a geometric-attention neural network to relate FC topology changes to brain activities. However, sliding window techniques are sensitive to window size, where suboptimal patterns can impair the detection of subtle brain state transitions.

The widespread success of recurrent neural networks (RNNs, Fig. 1 (a)) (Rumelhart et al., 1986), including long short-term memory (LSTM) (Hochreiter & Schmidhuber, 1997) and gated recurrent units (GRU) (Cho, 2014), in sequential modeling tasks such as natural language processing (NLP), has inspired numerous efforts to apply these architectures for characterizing brain dynamics (Li & Fan, 2018; 2019). Recently, state space models (SSMs) (as shown in Fig.1 (b, black solid box)) (Gu et al., 2021; 2022) have emerged as a powerful tool for capturing a system's behavior using hidden variables, or "states", marked as $s_t$ (i.e., $s(t)$), which effectively model temporal dependencies in sequential data with well-established theoretical properties. These models have gained significant attraction in fields like computer vision (CV) and NLP due to their ability to represent complex temporal patterns. A more inclusive literature survey can be found in the Appendix A.1.

**Relevant work of SSM on brain functional studies.** Motivated by the great success of SSM in CV and NLP applications, there are a number of learning-based SSMs proposed to understand the dynamic characteristics of functional activation, primarily applying these models to event-related (task-based) fMRI data analysis (Faisan et al., 2007; Hutchinson et al., 2009). Since these models sought to link each brain state to external stimuli (i.e., events), they are not well-suited for analyzing resting-state fMRI (rs-fMRI) data. To address this limitation, Suk et al. (2016) employed an auto-encoder model to learn the relationship between regional mean time series of BOLD signals and latent states and a hidden Markov model (HMM) to characterize the state transitions. However, the auto-encoder and HMM are trained separately in this work, which limits its overall efficiency. Additionally, this approach only focuses on capturing the dynamics of brain activity from BOLD signals, ignoring the crucial spatial structural information of the brain network. Meanwhile, Tu et al. (2019) proposed a linear SSM, leveraging a mean-field variational Bayesian approach, to infer causal-like effective connectivities from observed electroencephalography (EEG) and fMRI data. Due to the dynamic nature of FCs, however, SSM at the connectivity level only has limited power to uncover the complex relationship between evolving FCs and the underlying behavior/cognitive outcomes.

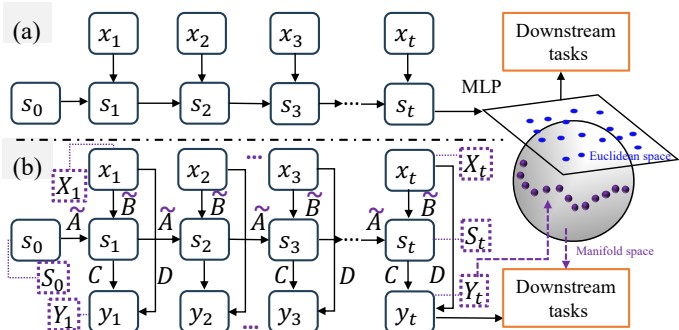

Figure 1: The architecture of RNNs (a) typically relies on a Multi-Layer Perceptron (MLP) to project the hidden state space into the output space, where various downstream tasks are then performed. These models operate entirely within Euclidean space. In contrast, vanilla SSMs (b, black solid box) incorporate two ODEs—the state equation (upper) and observation equation (lower)—which can directly perform downstream tasks through the inferred observed output, also within Euclidean space, focusing primarily on temporal information. Our proposed geometric deep model of SSM (b, purple dashed box) extends this approach by capturing both temporal and spatial information, operating on a manifold space.

**Our work.** The dynamic nature of complex system cannot be understood by thinking of the system as comprised of independent elements. Rather, an approach is needed to utilize knowledge about

the complex interactions within a system to understand the behavior of the system overall. In light of this, modeling the fluctuation of functional connectivities on the Riemannian manifold provides a holistic view of understanding how brain function emerges in cognition and behavior. In this paper, we integrate the power of geometric deep learning on Riemannian manifold and the mathematical insight of SSM to uncover the interplay between evolving brain states and observed neural activities. *First*, our method is *structural* in that we propose to learn intrinsic FC feature representations on the Riemannian manifold of SPD matrices, which allows us to take the whole-brain wiring patterns into account by considering each FC matrix as a manifold instance. *Second*, our method is *behavioral* in that we leverage the SSM to model temporal dynamics. As shown in Fig. 1 (b), SSMs operate through two core ordinary differential equations (ODEs)–the state equation and the observation equation–which describe the relationship between the input $x(t)$ (short for $x_t$) of the dynamic system and the system output $y(t)$ (short for $y_t$) at a given time $t$, mediated by a latent state $s(t)$ (short for $s_t$). Taken together, **our contribution** has three folds. (1) We present a novel geometric deep model by integrating state space model and manifold learning. By incorporating Riemannian geometry, our deep model provides an in-depth insight into system dynamics and state transitions, enhancing the model's ability to capture both temporal and spatial complexities in a data-driven manner. (2) We replace the Euclidean algebra of conventional SSMs with Riemannian geometric algebra (accompanied by theoretical analysis) to effectively capture the spatio-temporal information, which allows us to better handle irregular data structures and harness the geometric properties of SPD matrices. (3) We have significantly improved the computational efficiency compared to manifold-based deep models by using modern machine techniques such as geometric deep model (Sec. 3.1) and geometric-adaptive attention mechanism (Sec. 3.2).

We have applied our proposed method to two types of system dynamics: brain dynamics and action recognition (Bilinski & Bremond, 2015; Guo et al., 2013). While brain dynamics is our primary focus, action recognition serves as a validation task to assess the method's generalization performance across different domains. In the application of understanding brain dynamics, upon which we refer to as *GeoMind*, we have evaluated model performance on the large-scale human brain connectome (HBC) databases – one Human Connectome Project (Zhang et al., 2018) and four disease-related resting-state fMRI data: (1) Alzheimer's Disease Neuroimaging Initiative (ADNI) (Mueller et al., 2005), (2) Open Access Series of Imaging Studies (OASIS) (LaMontagne et al., 2019), (3) Parkinson's Progression Markers Initiative (PPMI) (Marek et al., 2011), and (4) the Autism Brain Imaging Data Exchange (ABIDE). For action recognition, we use three classic human action recognition (HAR) datasets including the Florence 3D Actions dataset (Seidenari et al., 2013), the HDM05 database (Müller et al., 2007), and the UTKinect-Action3D (UTK) dataset (Xia et al., 2012). Our *GeoMind* has achieved significant results across both brain dynamics and action recognition tasks, demonstrating its effectiveness and practicality. These applications on both neuroscience and computer vision highlight the scalability and robustness of our proposed approach in understanding complex spatio-temporal dynamics across diverse systems.

## 2 PRELIMINARY

### 2.1 STATE SPACE MODEL

The system dynamics typically formulate as the following state space model:
$$s'(t) = As(t) + Bx(t) \quad \text{and} \quad y(t) = Cs(t) + Dx(t) \tag{1}$$
where $s(t) \in \mathbb{R}^N$ indicates the current state, $A \in \mathbb{R}^{N \times N}$ denotes the transition matrix, $x(t) \in \mathbb{R}$ denotes the control input, $B \in \mathbb{R}^{N \times 1}$ represents the influence of control variables on state variables. $y(t) \in \mathbb{R}^M$ denotes the output of the system (it considers single-input and single-output conventions, i.e., $M = 1$), $C \in \mathbb{R}^{M \times N}$ represents the influence of the current state on output, $D \in \mathbb{R}^{M \times 1}$ (usually set as 0) denotes the influence of control variables on system output, as shown in Fig. 1 (b).

### 2.2 RIEMANNIAN GEOMETRY ALGEBRA

*Distance Metric.* Following the notation in (Chakraborty et al., 2018), we use $\mathcal{M}$ to represent the set of $N \times N$ SPD matrices ($X_{sym_N^+} \in \mathcal{M}$), and let $\mathbb{G}$ denotes the general linear group of $N \times N$ full-rank matrices. The group $\mathbb{G}$ acts on $X$ via the group action $g.X := gXg^\top$, where $g \in \mathbb{G}$. Furthermore, we employ the Stein metric (Cherian et al., 2011), defined as

$d(X, Y) = \sqrt{\log \det \left( \frac{X+Y}{2} \right) - \frac{1}{2} \log \det(XY)}$, to measure the distance between two SPD matrices ($X_{sym_N^+}, Y_{sym_N^+} \in \mathcal{M}$). This metric is notably more computationally efficient, as it circumvents the need for eigen decomposition.

*"Translation" Operation on Manifold $\mathcal{M}$.* Let $\mathbb{I}$ represent the set of all isometries on $\mathcal{M}$, meaning that for any $g \in \mathbb{I}$, the distance between points is preserved: $d(g \boldsymbol{.} X, g \boldsymbol{.} Y) = d(X, Y)$ for all $X, Y \in \mathcal{M}$. It is evident that $\mathbb{I}$ forms a group, and for any given $g \in \mathbb{I}$ and $X \in \mathcal{M}$, the mapping $g \boldsymbol{.} X \mapsto Y$, where $Y \in \mathcal{M}$, defines a group action. With the Stein metric, $\mathbb{I}$ corresponds to the general linear group $\mathbb{G}$. In this context, we focus on a subgroup of $\mathbb{G}$, specifically the orthogonal group $\mathbb{O}$, which consists of all $N \times N$ orthogonal matrices. For any $g \in \mathbb{O}$, the group action is defined as $g \boldsymbol{.} X := g X g^\top$. Since this group action preserves distances, it is referred to as a "translation" on the manifold, analogous to translations in Euclidean space, and is denoted by $\mathcal{T}_X(g) := g X g^\top$.

*Weighted Fréchet Mean (wFM) of Matrices on Manifold $\mathcal{M}$.* Given a set of matrices $\{X_n\}_{n=1}^N \subset \mathcal{M}$ and corresponding non-negative weights $\{w_n\}_{n=1}^N$ with $\sum_{n=1}^N w_n = 1$, the weighted Fréchet mean (wFM) is defined as the matrix $F^*$ that minimizes the weighted sum of squared distances to the elements in the set: $F^* = \underset{F}{\arg\min} \sum_{n=1}^N w_n d^2 (X_n, F)$. We assume that the matrices $X_n$ lie within a geodesic ball of an appropriate radius, ensuring the existence and uniqueness of the Fréchet mean. Henceforth, we denote the wFM of $X_n$ with weights $w_n$ as $\mathcal{F}(X_n, w_n)$.

*Convolution Operation on SPD manifold $\mathcal{M}$.* The SPD convolution operation of the $k^{th}(k = 1, \dots, K)$ network layer is depicted as

$$X_{i,j}^{(k)} = \sum_{u=0}^{\theta-1} \sum_{v=0}^{\theta-1} H_{u,v} X_{i+u,j+v}^{(k-1)} \tag{2}$$

where $H \in \mathbb{R}^{\theta \times \theta}$ is the convolutional kernel, $X_{i,j}^{(k)} \in \mathbb{R}^{(N-\theta+1) \times (N-\theta+1)}$ is the feature representation at matrix location $(i, j)$ of the $k^{th}(k = 1, \dots, K)$ network layer. Herein, if $H$ is a SPD matrix resulting in a SPD matrix $X^{(k)}$ (the proof is shown in the Appendix A.3). To maintain the SPD geometric structure during feature presentation learning, the SPD convolutional kernel $H$ is constructed by using multiplication of one matrix $Z \in \mathbb{R}^{(\theta \times \theta)}$, i.e., $H = Z^\top Z + \epsilon I$, where $\epsilon \to 0^+$, and $I$ is an identity matrix for guaranteeing that $H$ is dominantly diagonal. By doing so, we only need to learn the parameter $Z$, free of the constraint to ensure that the entire learning processing is implemented on the SPD manifold.

## 3 METHOD

### 3.1 GEOMETRY DEEP MODEL OF SSM

**Overview.** We propose a geometric deep model of SSM by extending the model design from Euclidean space to the manifold space. Unlike vanilla SSMs, where observations and hidden states evolve in Euclidean space, our approach models the data instance as a sequence of FC matrices over time (i.e., a series of SPD instances on manifold). Since the system input $X(t) \in \mathbb{R}^{N \times N}$ at time $t$ lies on the SPD manifold $\mathcal{M}$, we require the hidden state, $S(t)$, to be also represented as an SPD matrix. We further expect the system output, $Y(t)$, to follow the same SPD property, which allows us to capture the entire evolutionary state of the data on the Riemannian manifold space, preserving its inherent geometric structure.

**Problem formulation.** Following the Markov decision process (MDP) (Mnih et al., 2015), we introduce an "agent" $\mathcal{A}$ to control the evolution of states by calculating intrinsic control inputs $X(t)$. The objective is to ensure that the transformed input closely approximates the real input while embedding it within a high-dimensional manifold and preserving its original geometric structure. In this context, the agent $\mathcal{A}$ is trained to learn a stochastic policy that, at each step $k$, maps the history of previous interactions with the environment to a probability distribution over the actions at step $k$. At each step, the agent alternatively performs three key actions: (i) Updates the control input $X^{(k)}$ by imposing a convexity constraint on the weights $B$ to ensure the input becomes more aligned within the system. (ii) Captures the system dynamics $S^{(k)}$ by integrating a learnable transition matrix $A$. (iii) Updates the system state through a "translation" operation on the manifold. The main learning components of our approach can be summarized as follows:

*Internal state.* The agent maintains the current internal state $S^{(k)}$ that summarizes the representation of FC matrices inferred from the history state $S^{(k-1)}$ and the impact of control signals $X^{(k)}$ on the current system state $S^{(k)}$. The agent perceives the evolving environment (inferred current state $S^{(k)}$) by deciding how to act (for inducing geometric information on the FC matrices). After that, we can derive the system output $Y^{(k)}$ (i.e., observation) from the current hidden states by an observation equation. To do so, the internal state and system output can be formulated by a set of translation $\mathcal{T}$ and weighted Fréchet mean $\mathcal{F}$ (defined in Sec. 2.2) on the manifold $\mathcal{M}$:

$$
\begin{aligned}
S^{(k)} &= \mathcal{T}\left(\mathcal{F}\left(\{S^{(k-1)}\}, \{\widetilde{A}\}\right), \mathcal{F}\left(\{X^{(k)}\}, \{\widetilde{B}\}\right)\right) \\
Y^{(k)} &= \mathcal{T}\left(\mathcal{F}\left(\{S^{(k)}\}, \{C\}\right), \mathcal{F}\left(\{X^{(k)}\}, \{D\}\right)\right)
\end{aligned}
\tag{3}
$$

where $\widetilde{A}$ and $\widetilde{B}$ is discretized by $\widetilde{A} = \exp(\Delta A)$, $\widetilde{B} = (\Delta A^{-1}(\exp(\Delta A) - I) \cdot \Delta B$. In addition, $\Delta$ is the step size of discretization and $A, B, C, D$ are the learnable parameters. The whole workflow is shown in Fig. 1 (b, highlighted in purple).

*Actions.* At each step $k$, the inferred state influences the changes in the environmental variables. Specifically, the task/event, associated with the evolution of brain states $\{q|(1, ..., Q)\}$, is determined from a distribution parameterized by a softmax function applied to the system's output, i.e., $P(Q = q \mid \hat{y}^{(k)}) = \frac{\exp(w_q^\top \hat{y}^{(k)})}{\sum_{q'=1}^{Q} \exp(w_{q'}^\top \hat{y}^{(k)})}$, where $\hat{y}^{(k)}$ is the vectorized system output at step $k$, computed by logarithmic mapping $\hat{y}^{(k)} = \log(Y^{(k)}) = \Phi \log(\Lambda) \Phi^\top$. Note, $\log(\Lambda)$ is the diagonal matrix of eigenvalue logarithm, and $w_q$ represents the weight vector for specific brain task $q$. The softmax function calculates the probability of each class $q$ based on the system's output, yielding a probability distribution over the $Q$ classes (e.g., brain tasks/events/clinical outcomes underlying particular brain states).

*Rewards.* After executing the actions, the agent continuously influences the system's state evolution through feedback, which is typically quantified by minimizing the recognition error to maximize the overall benefit. Thus, we define the reward as: $\mathcal{L} = -\sum_{k=1}^{K} \sum_{q=1}^{Q} o_{kq} \log P(Q = q \mid \hat{y}^{(k)})$, where $o_{kq}$ is the one-hot encoded ground truth label for class $q$ at step $k$, with $o_{kq} = 1$ indicating that the inferred system output corresponds to the true label, and $o_{kq} = 0$ otherwise. The goal is to minimize this loss, driving the system towards more accurate predictions.

**Efficient geometric neural network of SSM.** To efficiently conduct the inference process, we can re-formulate the SSM (in Eq. 2) as a global convolution operation $\mathcal{K}$ (in Eq. 1) over time as follows:

$$
\mathcal{K} = \left(CB + D, CAB, \ldots, CA^{(k)}B, \ldots\right), \quad y = x * \mathcal{K}
\tag{4}
$$

The evolution of this formulation is described in Appendix A.2. In this context, we discrete this learning process of the agent into convolution operations on the manifold, based on Eq. 2. Thus, the multi-channel convolution operation on the manifold yields a multi-channel output as:

$$
X_{i,j}^{(k)} = \{X_{i,j}^{(k)}(r)\}_{r=1}^{R}, \quad X_{i,j}^{(k)}(r) = \sum_{l=0}^{L-1} \sum_{u=0}^{\theta-1} \sum_{v=0}^{\theta-1} \hat{\mathcal{K}}_{u,v}^{r,l} X_{i+u,j+v}^{(k-1)}(l),
\tag{5}
$$

where $R$ denotes the number of convolutional kernels, $L$ represents the channel number, and $\hat{\mathcal{K}} \in \mathbb{R}^{R \times L \times \theta \times \theta}$ is the multi-channel convolution kernels where each kernel $\mathcal{K}^{r,l}$ is an SPD matrices. In Eq. 5, $X^{(k)} \in \mathbb{R}^{R \times (N-\theta+1) \times (N-\theta+1)}$ denotes the output of the current layer and is also an SPD matrix (the proof is shown in the Appendix A.3). According to Eq. 4, we define $\hat{\mathcal{K}} = \mathcal{K}^\top \mathcal{K} + \epsilon I$, where learning $\mathcal{K}$ ensures the preservation of the SPD property. Next, we employ the elementwise operation $\exp(\cdot)$ operation as a non-linear activation function on the Riemannian algebra, ensuring the output remains an SPD matrix (see proof in Appendix A.4). The resulting SPD matrices are then normalized using the Frobenius norm to ensure bounded eigenvalues and maintain numerical stability. The model operates in convolutional mode for efficient, parallelizable training, processing the entire input sequence simultaneously. During autoregressive inference, it transitions to a recurrent mode (Eq. 3), enabling efficient step-by-step processing as inputs are received sequentially.

**Comparison between vanilla SSM and our geometric SSM.** It is worthwhile to noting that our formulation of system state update and observation equation through MDP offers a new insight of

learning mechanism in SSM, which is beyond the extension from Euclidean space to Riemannian manifold. As outlined in Eq. 3, we compute a weighted combination of prior information, where $S^{(k)}$ serves as the current state or token, and we transform $X^{(k)}$ using a "translation" operation. This process aggregates the information gathered at the current time step. Moreover, our geometric SSM integrates the power of MDP, enabling greater adaptability to diverse states, efficient decision-making, enhanced model interpretability, and scalability to complex dynamic systems. In addition to conventional SSMs, our update rules are highly nonlinear, taking into account both spatial structural information and temporal dynamics. Taken together, our framework demonstrates improved learning performance compared to vanilla SSMs by leveraging the geometric (covariance) structure and applying global convolutional operations on the manifold. We present the number of parameters and runtime for different models in Table 5.

### 3.2 GEOMETRIC-ADAPTIVE ATTENTION

To uncover the geometric pattern associated with the related diseases and brain tasks, we introduce a geometric-adaptive attention (GaA) module, which is bound to the SPD convolution kernel $\mathcal{K}$. In order to preserve the geometric structure information of the original input matrix to the greatest extent possible, GaA is designed to ensure that both the input and output matrices retain SPD properties while preserving their dimensionality. By doing so, we pad the edges of the output matrix with zeros of size $\theta - 1$ and introduce a small positive diagonal value to maintain the SPD properties (proved in Appendix A.5). The resulting geometric transformation is defined as:

$$\delta(\cdot) = \frac{\exp([X * \mathcal{K}])}{\max(\exp([X * \mathcal{K}]))} \tag{6}$$

where $[\cdot] = diag(\theta, X) = \begin{bmatrix} \theta & 0 \\ 0 & X \end{bmatrix}$ denotes SPD padding operation. This formulation is inspired by the standard sigmoid function, which maps the input values to the range [0,1], thereby preserving the SPD structure. Following this notion, we apply element-wise multiplication between the attention weights and the features to effectively capture system dynamics. This module leverages geometric properties to enhance the attention mechanism, enabling the model to adaptively capture both spatial and structural relationships within the inferred data. By incorporating geometric features from $X$, it extends traditional attention mechanisms, which typically operate in Euclidean space, into a manifold-aware framework. This transition leads to a more robust representation of the underlying data, especially when working with complex structures such as graphs or SPD matrices. The geometric-adaptive attention module enhances the model's focus on relevant patterns by accounting for both temporal and spatial dependencies in a principled geometric context, resulting in improved performance across tasks that involve intricate spatio-temporal relationships.

## 4 EXPERIMENTS

### 4.1 DATASET

We apply our method to two types of datasets including *human action recognition* (HAR) and *human brain connectome* (HBC), more detailed data information is shown in Table 3 and Appendix A.6.

**For HAR dataset.** We evaluate the performance of the proposed *GeoMind* on three widely-used HAR benchmarks: the Florence 3D Actions dataset (Seidenari et al., 2013), the HDM05 database (Müller et al., 2007), and the UTKinect-Action3D (UTK) dataset (Xia et al., 2012). The Florence 3D Actions dataset consists of 9 activities performed by 10 subjects, with each activity repeated 2 to 3 times, resulting in a total of 215 samples. The actions are captured by the motion of 15 skeletal joints. For the HDM05 dataset, we follow the protocol from (Wang et al., 2015), focusing on 14 action classes. This dataset contains 686 samples, each represented by 31 skeletal joints. Lastly, the UTKinect-Action3D dataset comprises 10 action classes. Each action was performed twice by 10 subjects, yielding a total of 199 samples.

**For HBC dataset.** We select one dataset of healthy young adults and four disease-related human brain datasets for evaluation: the HCP Working Memory (HCP-WM) (Zhang et al., 2018), ADNI (Mueller et al., 2005), OASIS (LaMontagne et al., 2019), PPMI (Marek et al., 2011), and ABIDE (Di Martino et al., 2014). We selected a total of 1,081 subjects from the HCP-WM dataset. The

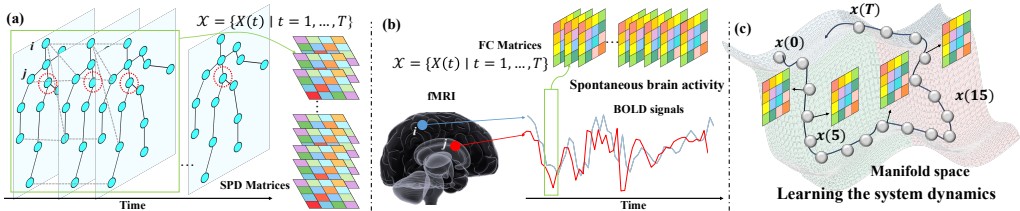

Figure 2: The construction of SPD matrices for HAR (a) and HBC (b) datasets. Leaning the system dynamics on manifold space as illustrated in (c).

working memory task included eight task events. Brain activity was parcellated into 360 regions based on the multi-modal parcellation from (Glasser et al., 2016). For the OASIS (924 subjects) and ABIDE (1,025 subjects) datasets, which are binary-class datasets, one class represents a disease group and the other represents healthy controls. In the ADNI dataset, subjects are categorized based on clinical outcomes into four distinct cognitive status groups. The PPMI dataset also consists of four classes. We employ Automated Anatomical Labeling (AAL) atlas (Tzourio-Mazoyer et al., 2002) (116 brain regions) on ADNI, PPMI, ABIDE datasets, while Destrieux atlas (Destrieux et al., 2010) (160 brain regions) are used in OASIS to verify the scalability of the models.

## 4.2 SPD MATRICES CONSTRUCTION

**For HAR dataset.** HAR datasets exhibit variability due to differences in action duration, complexity, the number of action classes, and the technology used for data capture. Therefore, we first apply a preprocessing step following (Paoletti et al., 2021) to obtain the SPD matrices. This step involves fixing the root joint at the hip center (red dashed circle in Fig. 2 (a)) and calculating the relative 3D positional differences for all other $N-1$ joints. For each timestamp $t = 1, \ldots, T$, we obtain a $3 \times (N-1)$-dimensional column vector $p(t)$ representing the relative displacements of the joints. Then, we compute covariance matrices using the method proposed in (Paoletti et al., 2021) to yield the SPD matrices. After that, we apply a sliding window technique to capture the dynamics over time, resulting in a sequence of SPD matrices $\mathcal{X} = \{X(t) \mid t = 1, \ldots, T\} \in \mathbb{R}^{T \times (3(N-1)) \times (3(N-1))}$, as illustrated in Fig. 2(a).

**For HBC dataset**. Assuming each fMRI scan has been processed into $N$ mean time courses of BOLD signals, each with $T$ time points (where $N$ represents the number of brain parcellations), we employ a sliding window technique to capture functional brain dynamics. Specifically, we construct a $N \times N$ correlation matrix at each time point $t$ ($t = 1, \ldots, T$) based on the BOLD signal within the sliding window, centered at time $t$. This results in a sequence of FC matrices encoding the functional dynamics for each scan, represented as $\mathcal{X} = \{X(t) \mid t = 1, \ldots, T\} \in \mathbb{R}^{T \times N \times N}$, in Fig. 2 (b).

## 4.3 COMPARISON METHODS AND EVALUATION METRICS

**For HAR dataset.** There are some popular methods for HAR, such as multi-part bag-of-pose (MBP) (Seidenari et al., 2013), Lie group (Vemulapalli et al., 2014), shape analysis on manifold (SAM) (Devanne et al., 2014), elastic function coding (EFC) (Anirudh et al., 2015), multi-instance multitask learning (MML) (Yang et al., 2016), Tensor Representation (TR) (Koniusz et al., 2016), LieNet(Hussein et al., 2013), SPGK (Wang et al., 2016), ST-NBNN (Weng et al., 2017), GR-GCN (Gao et al., 2019) and DMT-Net and F-DMT-Net (Zhang et al., 2020). We also include Bi-long short-term memory (Bi-LSTM) (Ben Tanfous et al., 2018) and pair-ware LSTM (P-LSTM) (Shahroudy et al., 2016).

**For HBC dataset.** We stratify the comparison methods for HBC into two groups: spatial and sequential models. *Spatial models* focus on capturing brain dynamics. Traditional GNNs like GCN (Kipf & Welling, 2016) and GIN (Xu et al., 2018) are included for their ability to handle structured data. Subgraph-based GNNs like Moment-GNN (Kanatsoulis & Ribeiro, 2023) focus on identifying local patterns, while expressive GNNs like GSN (Bouritsas et al., 2022) and GNN-AK (Zhao et al., 2021) enhance subgraph encoding for better expressivity. SPDNet (Dan et al., 2022b), a manifold-based model, is chosen for managing high-dimensional data. Plus, an MLP serves as a simple, generic baseline. *Sequential models* target temporal dynamics in BOLD signals. 1D-CNN captures temporal patterns, while RNN (Rumelhart et al., 1986) and LSTM (Hochreiter & Schmidhuber, 1997) handle sequential dependencies. MLP-Mixer (Tolstikhin et al., 2021) integrates both temporal and spatial information, and Transformer (TF) (Vaswani et al., 2017) captures global dependencies through attention. Mamba (Gu & Dao, 2023), vanilla SSM, is included for its ability to model

Table 1: Results on HAR dataset.

| Florence | | UTKinect | | HDM05 | |
|---|---|---|---|---|---|
| **Methods** | **Accuracy (%)** | **Methods** | **Accuracy (%)** | **Methods** | **Accuracy (%)** |
| MBP | 82.00 | Lie group | 97.10 | Lie group | 70.26 ± 2.89 |
| Lie group | 90.08 | EFC | 94.90 | LieNet | 75.78 ± 2.26 |
| SAM | 66.20 | SPGK | 97.40 | SPDNet | 61.45 ± 1.12 |
| EFC | 87.04 | ST-NBNN | 98.00 | P-LSTM | 73.42 ± 2.05 |
| MML | 89.67 | Bi-LSTM | 96.90 | DMT-Net | 81.52 ± 1.17 |
| TR | 95.47 | GR-GCN | 98.50 | F-DMT-Net | 85.30 ± 1.58 |
| *GeoMind* | **98.96** | *GeoMind* | **98.67** | *GeoMind* | **89.85 ± 1.86** |

system dynamics over time. Two dynamic-FC methods, STAGIN (Kim et al., 2021), NeuroGraph (Said et al., 2023). Three brain network analysis methods BrainGNN (Li et al., 2021), BNT (Kan et al., 2022), and ContrastPool (Xu et al., 2024). More details are shown in Appendix A.7.

**Evaluation metrics.** For the Florence and UTKinect datasets, we adopt the standard leave-one-actor-out validation protocol as outlined in (Gao et al., 2019). This method generates $Q$ classification accuracy values, which are averaged to produce the final accuracy score. For the HDM05 dataset, we follow the experimental setup from (Huang & Van Gool, 2017), conducting 10 random evaluations. In each evaluation, half of the samples from each class are randomly selected for training, with the remaining half used for testing. In all HBC experiments, we utilize a 10-fold cross-validation scheme, reporting accuracy (Acc), precision (Pre), and F1 score to provide a thorough evaluation of model performance across various datasets.

## 4.4 RESULTS ON HUMAN ACTION RECOGNITION (HAR)

Table 1 presents the numerical results for the HAR dataset, demonstrating that our method delivers competitive performance. The superiority of our method lies in its ability to simultaneously capture spatio-temporal correlations while preserving the geometric structure between joints.

*Remark 1.* Our method effectively captures higher-order correlations between the 3D coordinates of body joints and their temporal dynamics. Additionally, our method models the spatio-temporal co-occurrences of body joints using a tailored global convolution kernel, which helps mitigate the impact of noisy joints and enhances overall action recognition accuracy.

## 4.5 RESULTS ON HUMAN BRAIN CONNECTIVITY (HBC)

In this section, we explore the brain dynamics of health (task-based fMRI) and disease-related (resting-state fMRI) cohorts. **Firstly**, we conduct a task-based recognition experiment for HCP-WM dataset on fourteen methods, Table 2 (first column) shows the performance on different methods. Sequential models demonstrate a notable performance (pair-wise $t$-test, $p < 10^{-4}$) advantage over spatial models, with up to a 30% increase in accuracy. Our proposed *GeoMind* achieves the best overall performance.

*Remark 2.* One possible explanation, from *the perspective of machine learning*, the superior performance of sequential models over spatial models may be due to the stronger correlation between the dynamic nature of BOLD signals and cognitive tasks, compared to the static wiring topology of functional connectivities in healthy brains. *Biologically*, this difference stems from the design of task-based experiments, which target specific brain responses related to cognitive functions like attention and memory. These tasks increase brain activity, as reflected in the fluctuating dynamics of BOLD signals.

**Secondly**, we analyze the early diagnosis of neurodegenerative diseases using resting-state fMRI, focusing on Alzheimer's Disease (AD) and Parkinson's Disease (PD) due to the availability of large public datasets. Specifically, we assess the classification performance between cognitively normal (CN) individuals and those with neurodegenerative diseases (ND). In these experiments, spatial models perform slightly better than sequential models (difference is not statistically significant with $p = 0.37$). Our proposed *GeoMind* demonstrates a much better performance in these tasks (outperforms the $2^{nd}$-ranked method with a significance improvement at $p < 0.01$).

*Remark 3.* In contrast to task-based fMRI, which captures brain activity in response to specific tasks, resting-state fMRI measures spontaneous brain activity, reflecting intrinsic functional connectivity between brain regions. These fundamental differences in biological mechanisms explain why spatial models often achieve higher classification accuracy than sequential models in this context. Neurological impairments in ND may result from dysfunction rather than outright neuron loss (Palop et al., 2006). ND can thus be viewed as a disconnection syndrome, where large-scale brain networks are

Table 2: Evaluation performance for different methods across HBC datasets. The best performance is highlighted in bold, while the second-best is underlined.

| | Metric | HCP-WM | ADNI | OASIS | PPMI | ABIDE |
|---|---|---|---|---|---|---|
| **1D-CNN** | Acc | 96.71 ± 0.74 | 76.00 ± 6.45 | 88.75 ± 1.87 | 68.02 ± 10.75 | 68.87 ± 3.10 |
| | Pre | 96.73 ± 0.73 | 72.92 ± 14.98 | 87.23 ± 5.95 | 65.33 ± 13.50 | 70.79 ± 3.71 |
| | F1 | 96.71 ± 0.74 | 68.99 ± 9.60 | 84.93 ± 2.88 | 61.41 ± 13.42 | 67.93 ± 3.14 |
| **RNN** | Acc | 94.54 ± 0.97 | 75.20 ± 6.14 | 87.15 ± 2.31 | 56.55 ± 7.21 | 56.97 ± 3.20 |
| | Pre | 95.60 ± 0.95 | 69.66 ± 75.82 | 77.30 ± 5.55 | 45.15 ± 15.34 | 59.66 ± 5.53 |
| | F1 | 94.54 ± 0.97 | 68.90 ± 8.94 | 81.25 ± 3.30 | 43.14 ± 8.46 | 48.52 ± 5.82 |
| **LSTM** | Acc | 96.61 ± 0.30 | 77.60 ± 6.25 | 87.07 ± 2.32 | 64.21 ± 10.56 | 56.68 ± 3.03 |
| | Pre | 96.64 ± 0.29 | 76.11 ± 13.32 | 75.87 ± 4.04 | 57.86 ± 18.23 | 53.37 ± 14.74 |
| | F1 | 96.61 ± 0.30 | 72.48 ± 9.05 | 81.07 ± 2.09 | 56.25 ± 15.00 | 45.10 ± 5.31 |
| **Mixer** | Acc | 96.88 ± 0.65 | 77.20 ± 5.95 | 87.15 ± 2.20 | 66.12 ± 11.03 | 62.24 ± 2.26 |
| | Pre | 96.93 ± 0.63 | 77.87 ± 12.76 | 77.60 ± 4.52 | 63.27 ± 16.97 | 64.37 ± 4.72 |
| | F1 | 96.89 ± 0.64 | 72.09 ± 9.56 | 81.26 ± 3.04 | 58.64 ± 14.44 | 60.68 ± 5.24 |
| **TF** | Acc | 97.77 ± 0.48 | 79.20 ± 5.31 | 88.03 ± 1.49 | 70.43 ± 11.74 | 67.02 ± 4.57 |
| | Pre | 97.80 ± 0.47 | 78.53 ± 10.50 | 85.58 ± 5.17 | 66.59 ± 13.26 | 67.53 ± 4.85 |
| | F1 | 97.77 ± 0.48 | 75.39 ± 7.58 | 83.61 ± 2.90 | 64.68 ± 14.35 | 66.63 ± 4.77 |
| **Mamba** | Acc | 96.76 ± 0.86 | 74.40 ± 5.43 | 87.09 ± 0.75 | 67.93 ± 10.69 | 66.34 ± 0.27 |
| | Pre | 96.80 ± 0.84 | 67.78 ± 14.50 | 75.93 ± 0.23 | 66.40 ± 11.44 | 68.26 ± 0.17 |
| | F1 | 96.76 ± 0.86 | 66.98 ± 8.59 | 81.10 ± 0.23 | 59.11 ± 8.87 | 66.30 ± 1.24 |
| **GCN** | Acc | 72.69 ± 2.14 | 74.40 ± 3.67 | 88.01 ± 1.70 | 68.02 ± 11.57 | 67.11 ± 4.49 |
| | Pre | 73.28 ± 1.93 | 67.12 ± 12.30 | 84.86 ± 4.42 | 60.28 ± 18.09 | 67.76 ± 4.14 |
| | F1 | 72.72 ± 2.09 | 67.52 ± 5.87 | 84.20 ± 2.13 | 61.56 ± 15.25 | 66.88 ± 4.44 |
| **GIN** | Acc | 72.52 ± 2.41 | 76.40 ± 6.05 | 87.93 ± 2.52 | 70.33 ± 8.72 | 65.27 ± 3.86 |
| | Pre | 73.02 ± 2.57 | 69.75 ± 16.55 | 83.17 ± 6.22 | 66.64 ± 11.05 | 66.45 ± 4.36 |
| | F1 | 72.40 ± 2.53 | 69.61 ± 9.92 | 83.59 ± 3.84 | 64.84 ± 10.62 | 64.96 ± 3.88 |
| **GSN** | Acc | 79.99 ± 1.91 | 79.20 ± 4.66 | 88.69 ± 1.69 | 70.40 ± 12.48 | 67.02 ± 3.17 |
| | Pre | 80.28 ± 1.83 | 82.37 ± 4.81 | 86.22 ± 2.42 | 70.63 ± 14.00 | 68.30 ± 3.72 |
| | F1 | 79.92 ± 1.87 | 75.75 ± 4.92 | 86.54 ± 1.82 | 66.95 ± 13.64 | 66.38 ± 3.38 |
| **MGNN** | Acc | 74.70 ± 1.65 | 76.80 ± 3.92 | 88.73 ± 2.27 | 69.45 ± 10.37 | 64.97 ± 4.57 |
| | Pre | 75.86 ± 1.39 | 76.80 ± 9.67 | 87.99 ± 4.92 | 63.10 ± 15.32 | 66.03 ± 5.28 |
| | F1 | 74.63 ± 1.71 | 72.49 ± 6.08 | 85.16 ± 3.73 | 63.23 ± 13.29 | 63.45 ± 6.77 |
| **GNN-AK** | Acc | 59.48 ± 0.95 | 77.20 ± 6.21 | 88.05 ± 2.00 | 68.83 ± 7.70 | 61.75 ± 3.23 |
| | Pre | 61.97 ± 1.09 | 75.52 ± 13.41 | 86.38 ± 4.05 | 63.26 ± 11.75 | 64.71 ± 5.36 |
| | F1 | 59.32 ± 1.12 | 71.46 ± 9.81 | 83.88 ± 2.95 | 63.76 ± 9.01 | 58.10 ± 5.93 |
| **SPDNet** | Acc | 85.61 ± 1.01 | 78.50 ± 5.73 | 88.37 ± 2.14 | 66.02 ± 10.10 | 70.33 ± 3.03 |
| | Pre | 85.89 ± 1.05 | 65.04 ± 9.01 | 86.19 ± 5.45 | 42.92 ± 15.25 | 70.95 ± 3.04 |
| | F1 | 85.57 ± 1.04 | 61.91 ± 13.62 | 84.66 ± 2.76 | 40.14 ± 17.60 | 70.02 ± 2.99 |
| **MLP** | Acc | 83.54 ± 1.20 | 80.40 ± 4.54 | 89.26 ± 1.86 | 58.98 ± 10.94 | 68.77 ± 2.96 |
| | Pre | 84.18 ± 1.10 | 81.38 ± 5.55 | 88.72 ± 3.05 | 62.43 ± 13.15 | 69.39 ± 2.86 |
| | F1 | 83.56 ± 1.23 | 78.46 ± 4.99 | 86.47 ± 2.09 | 57.84 ± 11.82 | 68.67 ± 3.08 |
| **STAGIN** | Acc | 91.05 ± 0.90 | 74.00 ± 5.13 | 88.97 ± 1.81 | 67.75 ± 8.65 | 69.36 ± 2.23 |
| | Pre | 91.11 ± 0.90 | 63.49 ± 15.47 | **89.33 ± 1.69** | 59.93 ± 13.32 | 69.94 ± 2.35 |
| | F1 | 91.02 ± 0.90 | 65.50 ± 8.45 | 85.46 ± 2.89 | 60.22 ± 10.83 | 68.86 ± 2.35 |
| **NeuroGraph** | Acc | 67.97 ± 1.41 | 77.60 ± 4.07 | 89.06 ± 2.05 | **73.31 ± 10.64** | 60.97 ±2.00 |
| | Pre | 68.59 ± 1.17 | 76.19 ± 10.87 | 88.71 ± 3.15 | 67.98 ± 14.56 | 62.98 ± 5.28 |
| | F1 | 67.92 ± 1.33 | 73.52 ± 5.94 | 85.93 ± 2.32 | 68.63 ± 12.23 | 58.78 ± 4.26 |
| *GeoMind* | Acc | **98.29 ± 0.26** | **81.20 ± 2.27** | **89.60 ± 1.87** | 71.35 ± 10.26 | **70.97 ± 3.47** |
| | Pre | **98.18 ± 0.34** | **83.18 ± 4.19** | 87.38 ± 2.12 | **76.07 ± 7.33** | **72.29 ± 4.14** |
| | F1 | **98.16 ± 0.35** | **78.72 ± 2.63** | 87.34 ± 3.26 | **70.60 ± 9.73** | **71.04 ± 3.60** |

progressively disrupted by neuropathological processes (Chiesa et al., 2017). Evidence suggests that (1) brain function deteriorates years before cognitive decline and (2) the prodromal period can last decades before clinical diagnosis (Viola et al., 2015). Table 2 provide solid evidence for the potential of deep models in the early diagnosis of ND, with potential applications in clinical routine.

**Thirdly**, we analyze neuropsychiatric disorders using resting-state fMRI, focusing on Autism conditions in ABIDE dataset. Table 2 (last column) shows that spatial models slightly outperform sequential models. Herein, it is important to highlight the consistent top performance of SPDNet across all evaluation metrics, second only to our *GeoMind*. Both SPDNet and our *GeoMind* share two key methodological innovations: (1) preserving the geometry of FC matrices through manifold-based feature representation learning (as shown in Fig. 2 (c)), and (2) utilizing a spatio-temporal framework to capture dynamic patterns within evolving FC matrices. The exceptional performance of SPDNet and *GeoMind* indicates that the effective diagnosis of neuropsychiatric disorders may depend on robust spatio-temporal feature representation, grounded in solid mathematical foundations. This is further reinforced by the biological evidence discussed below.

Figure 3: Critical connections from geometric attention map on HBC datasets.

*Remark 4.* Autism and other neuropsychiatric disorders (such as Bipolar Disorder and Schizophrenia) are marked by atypical neural connectivity, with increased or decreased variability in BOLD signals, as well as altered neural dynamics that affect the timing and coordination of brain activity, impacting social and cognitive processing (Müller & Fishman, 2011; Uher et al., 2014; Rudie & Dapretto, 2013; Menon, 2011; Just et al., 2012). Since Autism affects both network topology and neural dynamics, a spatio-temporal approach is better suited for accurate diagnosis. In contrast, for early detection of ND, spatial models tend to outperform sequential models, as cognitive decline in ND is often associated with widespread neurodegeneration and disrupted network function. Ultimately, integrating disease-specific pathophysiological insights is essential for developing and interpreting effective diagnostic tools.

**Finally**, we evaluate the brain attention maps on the HBC datasets. Specifically, we extract the attention matrix $\delta(\cdot)$ of our GaA module (Sec. 3.2) to analyze the contributions of brain regions and their connections during working-memory tasks, as well as their involvement in the progression of AD, PD and Autism. To clarify, we select the top-20 connections (with high weight) from $\delta(\cdot)$ and map them back into brain, as shown in Fig. 3. For HCP-WM dataset, the critical connections are mainly located in the default mode network (DMN, highlighted in blue dashed circles) and central executive network (in orange dashed circle), implying that these regions are highly related to the working-memory tasks. In AD (OASIS and ADNI dataset), the primary symptoms—cognitive decline, memory loss, and behavioral changes—are well-documented. Our analysis reveals that the most significant brain connections are found within the DMN and the somatosensory cortex (in green dashed circles). This suggests that, in addition to memory degeneration, some patients may experience abnormal responses to tactile stimuli or disruptions in body part sensation as the disease progresses. These findings highlight the impact of AD on sensory processing and bodily awareness. For PD, our analysis highlights key connections in the sensorimotor regions (in red dashed circles), the frontal lobe (purple dashed circle), DMN, and the cerebellum (in black dashed circle). These findings suggest that while PD primarily affects motor function, likely due to cerebellar dysfunction, it may also impact cognitive and emotional functions, indicating a broader neurological involvement beyond just motor control. For Autism, we also observe the responses of temporal lobe (in brown dashed circle) and visual region (in yellow dashed circle), implying that this disease is closely associated with challenges in language processing, motor coordination and social interaction. Though not cast in stone, most of the identified brain regions are aligned with current clinical findings.

*Remark 5.* Although different diseases exhibit significant variations (neuropsychiatric disorders and neurodegenerative diseases), there are consistent patterns across certain neurodegenerative diseases, such as AD and PD. From our experimental results, the attention mechanism we designed shows the potential in uncovering the underlying mechanisms and progression pathways common to these diseases. This mechanism could offer valuable insights into both the distinct and shared aspects of different disease conditions, aiding in the exploration of their pathogenesis.

## 5 CONCLUSION

This work presents a geometric deep model of SSM, *GeoMind*, for understanding behavior/cognition through deciphering brain dynamics. In line with theoretical analysis, our method integrates the principles of *geometric deep learning* and efficient feature representation learning on non-Euclidean data, specifically designed for learning on sequential data with inherent topological connections. We have achieved promising experimental results on human connectome data as well as human action recognition, indicating great applicability in real-world data for neuroscience and computer vision.

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

# A APPENDIX

## A.1 LITERATURE SURVEY

**RNN and its variants on manifold to neuroimaging application.** Recurrent neural networks (RNNs) have been reformulated as ordinary differential equations (ODEs) with continuous-time hidden states, as highlighted by LTCNet (Hasani et al., 2021). These models serve as effective algorithms for modeling time series data and are widely utilized across medical, industrial, and business domains. For instance, Cai et al. (2023) has demonstrated its potential for brain state recognition and Han et al. (2024) achieves continuous modeling of dynamic brain signals using ODEs. Furthermore, the survey proposed by Niu et al. (2024) provides a comprehensive overview of ODE applications in the field of medical imaging, showcasing their practicality and impact in this domain. Following this, several manifold-based RNN models have emerged. For instance, Chakraborty et al. (2018) introduced a statistical recurrent model defined on the manifold of symmetric positive definite (SPD) matrices and evaluated its diagnostic potential for neuroimaging applications. This approach underscores the effectiveness of utilizing manifold-based techniques to enhance the performance of RNNs in complex medical contexts. The RNN model formulated on Riemannian manifolds Jeong et al. (2021) is robustly supported by mathematical theory, as it utilizes covariance information to dynamically model time-series data (Jeong et al., 2023). This capability allows it to capture richer and more subtle representations within a higher-dimensional latent space. Such an approach is particularly effective in modeling complex data structures, such as capturing the functional dynamics (Dan et al., 2022a; Huang et al., 2021), where the relationships among data points are inherently geometric. By operating within the manifold framework, these models adeptly accommodate the intricacies of underlying data distributions, thereby enhancing both interpretability and predictive performance.

RNNs and their variants, while widely used for sequential modeling tasks, have notable limitations that affect their performance in complex, dynamic systems. One of the key challenges is that RNNs implicitly learn sequential patterns and temporal dependencies, without explicitly modeling the underlying dynamics. This implicit nature makes RNNs harder to interpret, often turning them into "black-box" models where the relationships between input variables and predicted outcomes can be obscured, limiting their utility in scenarios requiring high interpretability. Although advancements like LTCNet (Hasani et al., 2021) have improved the interpretability of RNNs by framing them as an ODE, these models primarily focus on the dynamics of the hidden states and inputs (as shown in Fig. 1 (a)). However, they failed to consider observation equations (but usually use MLP to fit the observations), which describe the relationship between hidden states and observed data. This formulation reduces their ability to fully model the observable aspects of a system, resulting in an incomplete picture of the system's dynamics and limiting their explanatory power.

**SSM to neuroimaging application.** State Space Models (SSMs) explicitly model temporal dynamics through latent variables governed by two key ODEs: the state equation, which captures the evolution of the hidden state over time, and the observation equation, which relates the latent state to observable data. This structured, ODE-based framework allows SSMs to offer a clearer understanding of how systems evolve and provides a higher level of interpretability compared to RNNs. This makes SSMs particularly valuable in domains requiring an understanding of underlying system dynamics, such as medical diagnostics and time-series forecasting. In contrast to RNNs and their variants (e.g., LSTMs, GRUs), which often operate as "black boxes," SSMs like Kalman Filters (Kalman, 1960) have well-established theoretical properties. These properties typically include convergence and stability, providing a solid mathematical foundation that is difficult to guarantee with more complex RNN architectures. RNNs, especially deeper ones, can suffer from issues like vanishing or exploding gradients, which affect training stability and interoperability. SSMs also naturally incorporate probabilistic structures, allowing them to effectively handle noisy or uncertain data. This is particularly advantageous in low Signal-to-Noise Ratio (SNR) datasets, such as fMRI (Wu et al., 2019) and Electroencephalogram (EEG) data (Plub-in & Songsiri, 2018), where the ability to account for noise and uncertainty is critical. In light of these performance advantages, only a few manifold-based SSMs have been developed. For instance, Chikuse (2006) explores the modeling of time series observations in state-space forms defined on Stiefel and Grassmann manifolds. This approach utilizes Bayesian methods to estimate state matrices by calculating posterior modes, effectively integrating geometric constraints with probabilistic inference. However, while

Bayesian methods excel in handling uncertainty, they often face limitations in scalability, inference speed, and flexibility compared to deep learning models, which offer more efficient and powerful representation capabilities for large-scale data.

In this context, the introduction of deep geometric SSMs aims to combine the representational power of deep neural networks with the interpretability and structured dynamics inherent in traditional SSMs. By incorporating the geometric properties of manifold-based modeling, these models adeptly capture the intrinsic structure of the data, which is crucial for accurately representing complex relationships in high-dimensional datasets, such as those found in brain imaging. This combination not only enhances interpretability but also allows for a more nuanced understanding of the underlying dynamics, ultimately improving the efficacy of the modeling process.

## A.2 SSM TO CONVOLUTION OPERATION

$$
\begin{aligned}
s^0 &= Bx^0 \\
y^0 &= Cs^0 + Dx^0 = (CB + D)x^0 \\
s^1 &= As_0 + Bx^1 = ABx^0 + Bx^1 \\
y^1 &= Cs^1 + Dx^1 = C\left(ABx^0 + Bx^1\right) + Dx^1 = CABx^0 + (CB + D)x^1 \\
s^2 &= As^1 + Bx^2 = A\left(ABx^0 + Bx^1\right) + Bx^2 = A^2Bx^0 + ABx^1 + Bx^2 \\
y^2 &= Cs^2 + Dx^2 = C\left(A^2Bx^0 + ABx^1 + Bx^2\right) + Dx^2 = CA^2Bx^0 + CABx^1 + (CB + D)x^2 \\
y^k &= CA^kBx^0 + CA^{k-1}Bx^1 + \cdots + CABx^{k-1} + (CB + D)x^k \\
&\Rightarrow \mathcal{K} = \left(CB + D, CAB, \ldots, CA^kB, \ldots\right) \\
&\Rightarrow y = x * \mathcal{K}
\end{aligned}
\tag{7}
$$

Here, we abbreviate $x^{(k)}, s^{(k)}, y^{(k)}, A^{(k)}$ as $x^k, s^k, y^k, A^k$ for simplicity.

## A.3 SPD CONVOLUTION OPERATION

*Proof.* Since $H$ is SPD, it can be decomposed as follows:

$$
H = ZZ^\top,
\tag{8}
$$

where $Z = [z_1, z_2, \ldots, z_\theta]$ is a matrix of full rank. The convolutional result of an SPD representation matrix $X \in \mathbb{R}^{N \times N}$ can then be expressed as:

$$
O = X * H = X * (ZZ^\top),
\tag{9}
$$

$$
\Rightarrow X * (z_1 z_1^\top) + \cdots + X * (z_\theta z_\theta^\top),
\tag{10}
$$

$$
\Rightarrow X * z_1 * z_1^\top + \cdots + X * z_\theta * z_\theta^\top,
\tag{11}
$$

where the transition from Eq. 10 to Eq. 11 uses the property of separable convolution. Suppose $z_i = [m_{i1}, m_{i2}, \ldots, m_{i\theta}]^\top$, for $i = 1, 2, \ldots, \theta$. The convolution between $X$ and $z_i$ can be written as:

$$
X * z_i = P_{z_i}X, \quad X * z_i^\top = XP_{z_i}^\top,
\tag{12}
$$

where $P_{z_i} \in \mathbb{R}^{(M-N+1) \times M}$ and

$$
G_{z_i} = \begin{bmatrix}
m_{i1} & m_{i2} & \cdots & m_{iN} & 0 & 0 & \cdots \\
0 & m_{i1} & m_{i2} & \cdots & m_{iN} & 0 & \cdots \\
0 & 0 & m_{i1} & m_{i2} & \cdots & m_{iN} & \cdots \\
\vdots & \vdots & \vdots & \vdots & \vdots & \vdots & \vdots \\
0 & 0 & \cdots & 0 & m_{i1} & m_{i2} & \cdots & m_{iN}
\end{bmatrix}.
\tag{13}
$$

Thus, the following equations hold:

$$
X * z_i * z_i^\top = P_{z_i}XP_{z_i}^\top,
\tag{14}
$$

and

$$
O = X * Z = P_{z_1}XP_{z_1}^\top + \cdots + P_{z_\theta}XP_{z_\theta}^\top.
\tag{15}
$$

Since the rank of $P_{z_i}$ equals $M - N + 1$, the matrix $P_{z_i} X P_{z_i}^\top$ is also SPD. Therefore, for any $q \in \mathbb{R}^M$ where $q \neq 0$, we have:

$$q^\top O q = \sum_{i=1}^{\theta} q^\top P_{z_i} X P_{z_i}^\top q > 0. \tag{16}$$

Hence, $O$ is an SPD matrix.

Furthermore, the $k$-th channel of $X$ can be written as:

$$X^{(k)} = \sum_{l=1}^{L} X^{(l)} * H^{(k,l)}, \tag{17}$$

where $X^{(l)}$ denotes the $l$-th channel of the input descriptor. Since $X^{(l)}$ and $H^{(k,l)}$ are SPD matrices, and according to the above proof, $X^{(l)}$ is also an SPD matrix. Therefore, $X^{(k)}$ is a multi-channel SPD matrix.

### A.4 SPD $\exp(\cdot)$ OPERATION

*Proof:* Since $X$ is symmetric, we know that for any integer $k$, the powers $X^k$ are also symmetric. The matrix exponential of $X$ is defined by the following power series:

$$\exp(X) = \sum_{k=0}^{\infty} \frac{X^k}{k!}. \tag{18}$$

Each term in this series involves a symmetric matrix $X^k$, and the sum of symmetric matrices remains symmetric. Therefore, $\exp(X)$ is symmetric.

Since $X$ is symmetric, it can be diagonalized as: $X = Q \Lambda Q^\top$, where $Q$ is an orthogonal matrix (i.e., $Q^\top Q = I$) and $\Lambda$ is a diagonal matrix containing the eigenvalues $\lambda_1, \lambda_2, \ldots, \lambda_n$ of $X$. Because $X$ is positive definite, all eigenvalues $\lambda_i$ are positive, i.e., $\lambda_i > 0$ for all $i$.

The matrix exponential $\exp(X)$ is then given by:

$$\exp(X) = Q \exp(\Lambda) Q^\top, \tag{19}$$

where $\exp(\Lambda)$ is the diagonal matrix with entries $\exp(\lambda_1), \exp(\lambda_2), \ldots, \exp(\lambda_n)$. Since the exponential function satisfies $\exp(\lambda_i) > 0$ for all $\lambda_i \in \mathbb{R}$, each eigenvalue of $\exp(X)$ is positive. Thus, $\exp(X)$ has strictly positive eigenvalues, and since it is symmetric, it is also positive definite.

### A.5 SPD PADDING $[\cdot]$ OPERATION

Given a SPD matrices $X \in Sym_N^+$ and a small positive value $\theta$, the assemble matrix $Y = diag(\theta, X) = \begin{bmatrix} \theta & 0 \\ 0 & X \end{bmatrix}$ is a SPD matrix.

*Proof:* First, $Y$ is a symmetric, since $Y^\top = \begin{bmatrix} \theta & 0 \\ 0 & X \end{bmatrix}^\top = \begin{bmatrix} \theta & 0 \\ 0 & X \end{bmatrix} = Y$. Then, to show that $Y$ is positive definite, we need to verify that for any non-zero vector $z = \begin{bmatrix} z_1 \\ z_2 \end{bmatrix} \in \mathbb{R}^{N+1}$, the quadratic form $z^\top Y z$ is strictly positive.

We compute the quadratic form:

$$z^\top Y z = \begin{bmatrix} z_1 & z_2^\top \end{bmatrix} \begin{bmatrix} \theta & 0 \\ 0 & X \end{bmatrix} \begin{bmatrix} z_1 \\ z_2 \end{bmatrix} = z_1^2 \theta + z_2^\top X z_2. \tag{20}$$

Since $\theta > 0$, the term $z_1^2 \theta \geq 0$, and it is strictly positive if $z_1 \neq 0$.

Furthermore, since $X \in Sym_N^+$, $X$ is positive definite, meaning $z_2^\top X z_2 > 0$ for any non-zero $z_2 \in \mathbb{R}^N$.

Thus, for any non-zero vector $z = \begin{bmatrix} z_1 \\ z_2 \end{bmatrix}$, we have:

$$z^\top Y z = z_1^2 \theta + z_2^\top X z_2 > 0. \tag{21}$$

which proves $Y \in Sym_{N+1}^+$ is a SPD matrix.

## A.6 DATASET

Table 3: The summarization of the HAR and HBC datasets.

| Dataset | # of sequences | # of classes | mean of lengths | # of joints/ROIs |
|---|---|---|---|---|
| UTKinect | 199 | 10 | 29 | 20 |
| Florece 3D Actions | 215 | 9 | 19 | 15 |
| HDM05 | 686 | 14 | 248 | 31 |
| HCP-WM | 17,296 | 8 | 39 | 360 |
| ADNI | 250 | 5 | 177 | 116 |
| OASIS | 1,247 | 2 | 390 | 160 |
| PPMI | 209 | 4 | 198 | 116 |
| ABIDE | 1,025 | 2 | 200 | 116 |

**For HAR dataset.** We evaluate the performance of the proposed *GeoMind* on three benchmark HAR datasets: the Florence 3D Actions dataset (Seidenari et al., 2013), the HDM05 database (Müller et al., 2007), and the UTKinect-Action3D (UTK) dataset (Xia et al., 2012). The Florence 3D Actions dataset includes 9 activities (*answer phone, bow, clap, drink, read watch, sit down, stand up, tie lace, wave*), performed by 10 subjects, with each activity repeated 2 to 3 times, resulting in a total of 215 samples. These actions are represented by the motion of 15 skeletal joints. For the HDM05 dataset, we follow the protocol outlined in (Wang et al., 2015), selecting 14 action classes (*clap above head, deposit floor, elbow to knee, grab high, hop both legs, jog, kick forward, lie down on floor, rotate both arms backward, sit down chair, sneak, squat, stand up, throw basketball*). The sequences, captured using VICON cameras, result in 686 samples, each represented by 31 skeletal joints—significantly more than in the Florence dataset. The increased number of joints and higher intra-class variability make this dataset particularly challenging. Finally, the UTKinect-Action3D dataset consists of 10 action classes (*carry, clap hands, pick up, pull, push, sit down, stand up, throw, walk, wave hands*), captured using a stationary Microsoft Kinect camera. Each action was performed twice by 10 subjects, yielding 199 samples in total.

**For HBC dataset.** We select one dataset of healthy young adults and four disease-related human brain datasets for evaluation: the Human Connectome Project-Young Adult Working Memory (HCP-WM) (Zhang et al., 2018), Alzheimer's Disease Neuroimaging Initiative (ADNI) (Mueller et al., 2005), Open Access Series of Imaging Studies (OASIS) (LaMontagne et al., 2019), Parkinson's Progression Markers Initiative (PPMI) (Marek et al., 2011), and the Autism Brain Imaging Data Exchange (ABIDE). We selected a total of 1,081 subjects from the HCP-WM dataset. The working memory task included both 2-back and 0-back conditions, with stimuli featuring images of bodies, places, faces, and tools, interspersed with fixation periods. The specific task events are: 2bk-body, 0bk-face, 2bk-tool, 0bk-body, 0bk-place, 2bk-face, 0bk-tool, and 2bk-place. Brain activity was parcellated into 360 regions based on the multi-modal parcellation from (Glasser et al., 2016). For the OASIS (924 subjects) and ABIDE (1,025 subjects) datasets, which are binary-class datasets, one class represents a disease group and the other represent healthy controls. In the ADNI dataset, subjects are categorized based on clinical outcomes into distinct cognitive status groups: cognitively normal (CN), subjective memory concern (SMC), early-stage mild cognitive impairment (EMCI), late-stage mild cognitive impairment (LMCI), and Alzheimer's Disease (AD). For population analysis, we group CN, SMC, and EMCI into a "CN-like" group, while LMCI and AD form the "AD-like" group. This grouping enables a detailed analysis of cognitive decline and disease progression. The PPMI dataset consists of four classes: normal control, scans without evidence of dopaminergic deficit (SWEDD), prodromal Parkinson's disease, and Parkinson's disease (PD). This classification supports the study of different stages of Parkinson's progression. We employ Automated Anatomical Labeling (AAL) atlas (Tzourio-Mazoyer et al., 2002) (116 brain regions) on ADNI, PPMI, ABIDE datasets, while Destrieux atlas (Destrieux et al., 2010) (160 brain regions) are used in OASIS to verify the scalability of the models.

## A.7 COMPARSION METHODS AND EXPERIMENTAL RESULTS

We roughly summarize the comparison methods for HBC into two categories: spatial models and sequential models.

**Spatial models.** The spatial models are essential for understanding brain dynamics. Traditional GNNs like graph convolutional network (GCN) (Kipf & Welling, 2016) and graph isomorphism network (GIN) (Xu et al., 2018) are selected for their ability to effectively capture diffusion patterns and isomorphism encoding in structured data. Subgraph-based GNNs, such as Moment-GNN (Kanatsoulis & Ribeiro, 2023), emphasize subgraph structures, enabling the identification of localized patterns that might be overlooked by traditional GNNs. Expressive GNNs, including graph substructure network (GSN) (Bouritsas et al., 2022) and GNNAsKernel (GNN-AK) (Zhao et al., 2021), are chosen for their enhanced expressivity through subgraph isomorphism counting and local subgraph encoding, which could be crucial for distinguishing subtle differences in complex systems.

A manifold-based model like the symmetric positive definite network (SPDNet) (Dan et al., 2022b) is adopted for its ability to manage high-dimensional manifold data, making it suitable for more complicated datasets.

Two graph-based brain network analysis models for disease diagnosis, BrainGNN (Li et al., 2021), an interpretable brain graph neural network for fMRI analysis, and ContrastPool (Xu et al., 2024), a contrastive dual-attention block and a differentiable graph pooling method.

Additionally, a traditional multi-layer perceptron (MLP) serves as a model due to its efficiency and versatility across various domains.

For all spatial models, following the optimal settings described in (Said et al., 2023), we use the vectorized static functional connectivity (FC) as graph embeddings and the static FC matrices ($N \times N$) as adjacency matrices, where only the top 10% of edges are retained through thresholding to ensure sparsity. The input of SPDNet is the original $N \times N$ FC matrices.

For dynamic-FC models (STAGIN (Kim et al., 2021) and NeuroGraph (Said et al., 2023), the thresholded dynamic FC matrices serve as the graph, NeuroGraph serve the vectorized FC as the embedding and STAGIN incorporates BOLD signals as part of the embedding, alongside its unique embedding construction method. For our *GeoMind*, we use the dynamic FC matrices as the input, resulting in $T \times N \times N$ matrices.

**Sequential models.** The sequential models are selected for analyzing temporal dynamics in time-series BOLD signals. 1D-CNN is chosen for its ability to capture temporal patterns through convolutional operations. RNN (Rumelhart et al., 1986) and LSTM (Hochreiter & Schmidhuber, 1997) are included for their proficiency in modeling sequential data and capturing long-range dependencies. MLP-Mixer (Tolstikhin et al., 2021) is selected for its capability to mix both temporal and spatial features, offering a comprehensive view by integrating information across different dimensions. Transformer (Vaswani et al., 2017) is chosen for its powerful attention mechanisms, which allow it to capture global dependencies in sequential data. Brain network transformer (BNT) (Kan et al., 2022) is a tailored approach specifically designed for brain network analysis. Lastly, the state-space model (SSM), represented by Mamba (Gu & Dao, 2023), is selected for its advanced state-space modeling abilities that effectively capture system dynamics over time.

For the sequential models, the inputs are the BOLD signals ($N \times T$).

Note, the inputs for all comparison methods align with the recent work presented in (Ding et al., 2024), ensuring fairness in the evaluation process.

We further conducted experiments using three brain network analysis models on disease-based datasets, including ADNI, OASIS, PPMI, and ABIDE. The diagnostic accuracies of 10-fold cross-validation are presented in Table 4. It is clear that our *GeoMind* consistently outperforms all the compared methods.

Table 4: Diagnostic accuracies on three popular brain network analysis models.

|  | ADNI | OASIS | PPMI | ABIDE |
|---|---|---|---|---|
| BrainGNN | 76.57 ± 10.01 | 86.07 ± 5.71 | 67.88 ± 10.32 | 62.24 ± 4.44 |
| BNT | 79.68 ± 6.15 | 86.07 ± 3.19 | 64.55 ± 16.80 | 69.99 ± 5.37 |
| ContrastPool | 80.08 ± 5.01 | 89.02 ± 4.22 | 69.78 ± 7.36 | 70.72 ± 3.45 |

## A.8 INFERENCE TIME AND THE NUMBER OF PARAMETERS

We summarize the inference time and the number of parameters of each mode on HCP-WM dataset ($N = 360, T = 39$), all the experiments are conducted on NVIDIA RTX 6000Ada GPUs. We can observe that our method efficiently utilized the parameters compared to most counterpart methods. Compared to Mamba (vanilla SSM), our method requires more time in the final step due to the logarithmic mapping, which involves the computationally expensive Singular Value Decomposition (SVD). However, it is more efficient than SPDNet (a manifold-based model), as we leverage convolution operations. As a result, the overall computational cost remains manageable.

Table 5: Model inference time (ms/item) and the number of parameters (M) comparison across various architectures on HCP-WM dataset.

|           | GCN  | GIN  | GSN   | MGNN | GNN-AK | SPDNet     | MLP     | 1D-CNN  |
|-----------|------|------|-------|------|--------|------------|---------|---------|
| Time (ms) | 2.29 | 2.28 | 3.40  | 2.23 | 38.18  | 27.05      | 2.67    | 0.93    |
| Para (M)  | 1.79 | 3.89 | 0.92  | 4.94 | 290.3  | 0.19       | 66.9    | 2.22    |

|           | RNN  | LSTM  | Mixer | TF    | Mamba | NeuroGraph | STAGIN  | *GeoMind* |
|-----------|------|-------|-------|-------|-------|------------|---------|-----------|
| Time (ms) | 0.87 | 0.91  | 0.91  | 1.21  | 0.33  | 39.79      | 20.92   | 2.51      |
| Para (M)  | 1.19 | 14.45 | 6.78  | 12.98 | 27.05 | 0.29       | 1.17    | 14.60     |

More detailed information is shown in `https://anonymous.4open.science/r/GeoMind-12E8/`.

## A.9 ABLATION STUDY

We perform ablation studies to investigate the effects of sliding window size and the contribution of the proposed GaA module in the underlying *GeoMind* network architecture. For sliding window size, the experiments are performed on the PPMI dataset. For the evaluation without the GaA module, we conduct experiments on all datasets, with the sliding window size fixed at 15. The numerical results from 10-fold cross-validation are presented in Table 6.

Table 6: Ablation studies in terms of sliding window size and the contribution of GaA module in the underlying *GeoMind* network architecture.

| Window size | 15               | 25                | 35               | 45               | 55               |
|-------------|------------------|-------------------|------------------|------------------|------------------|
| Acc         | 71.35 ± 10.26    | 70.83 ± 15.74     | 71.69 ± 10.10    | 72.01 ± 8.51     | 71.01 ± 14.23    |
| Pre         | 76.07 ± 7.33     | 74.72 ± 10.00     | 73.54 ± 6.50     | 71.73 ± 7.56     | 71.00 ± 7.89     |
| F1          | 70.60 ± 9.73     | 71.71 ± 7.29      | 70.72 ± 3.90     | 68.56 ± 6.74     | 67.67 ± 7.81     |

| w/o GaA | HCP-WM        | ADNI         | OASIS        | PPMI         | ABIDE        |
|---------|---------------|--------------|--------------|--------------|--------------|
| Acc     | 97.25 ± 0.65  | 79.60 ± 2.80 | 89.26 ± 2.29 | 70.97 ± 8.02 | 69.75 ± 2.70 |
| Pre     | 97.29 ± 0.64  | 80.51 ± 4.92 | 87.37 ± 5.68 | 73.53 ± 8.93 | 69.90 ± 1.68 |
| F1      | 97.24 ± 0.66  | 76.86 ± 3.78 | 86.49 ± 3.52 | 67.34 ± 8.66 | 69.66 ± 1.24 |

We can observe that *GeoMind* demonstrates relative insensitivity to window size, with optimal performance observed at moderate values, typically within the range of 25 to 35. This robustness can be attributed to *GeoMind*'s reliance on the SSM module to capture dynamic temporal characteristics. Additionally, the proposed GaA module is an essential component of the network architecture, contributing significantly to its overall performance.

## A.10 DISCUSSION

We expect our manifold-based deep model to facilitate our understanding on brain behavior in the following ways.

*(1) Enhance the prediction accuracy.* A plethora of neuroscience findings indicate that fluctuation of functional connectivities exhibits self-organized spatial-temporal patterns. Following this notion,

we conceptualize that well-defined mathematical modeling of intrinsic data geometry of evolving functional connectivity (FC) matrices might be the gateway to enhance prediction accuracy. Our experiments have shown that respecting the intrinsic data geometry in method development leads to significantly higher prediction accuracy for cognitive states, as demonstrated in Table 2.

*(2) Enhance the model explainability.* We train the deep model to parameterize the transition of FC matrices on the Riemannian manifold (Eq. 4 and 5). By doing so, we are able to analyze the temporal behaviors with respect to each cognitive state using post-hoc complex system approaches such as dynamic mode decomposition, stability analysis.

*(3) Provide a high-order geometric attention mechanism that is beyond node-wise or link-wise focal patterns.* Conventional methods often employ attention components for each region or link in the brain network separately, thus lacking the high-order attention maps associated with neural circuits (i.e., a set of links representing a sub-network). In contrast, the geometric attention mechanism (Eq. 6) in our method operates on the Riemannian manifold, taking the entire brain network into account. As shown in Fig. 3, our method has identified not only links but also sub-networks relevant to cognitive states and disease outcomes.

