# OpenReview forum: "GeoMind: A Geometric Neural Network of State Space Model for Understanding Brain Dynamics on Riemannian Manifold"
_ICLR.cc/2025/Conference — ICLR 2025 Conference Withdrawn Submission_

### Official Review · Reviewer_QfXA · 2024-11-02

**Soundness:** 3
**Presentation:** 3
**Contribution:** 3
**Rating:** 6
**Confidence:** 4

**Summary:**

The paper introduces GeoMind, a geometric neural network designed to analyze brain state dynamics using state space models within a Riemannian manifold of symmetric positive definite (SPD) matrices. By leveraging the geometric properties of functional connectivities, GeoMind effectively uncovers evolving brain states from task-based fMRI data, showing promise for early diagnosis of neurological conditions such as Alzheimer’s, Parkinson’s, and Autism. Additionally, the model demonstrates versatility by achieving strong performance in human action recognition across multiple benchmark datasets.

**Strengths:**

The paper presents several strengths while also having notable limitations. One significant strength is the introduction of GeoMind, a geometric neural network that employs Riemannian manifolds and symmetric positive definite (SPD) matrices to analyze brain state dynamics. This innovative approach provides a unique framework for modeling complex neural activities, with promising results in identifying specific brain states from task-based fMRI data. Additionally, the model shows potential for early diagnosis of neurodegenerative diseases such as Alzheimer’s, Parkinson’s, and Autism, which underscores its relevance in clinical applications.

**Weaknesses:**

The paper has several limitations that affect its overall contribution. While it compares GeoMind with Mamba, it does not achieve state-of-the-art performance on key datasets such as ADNI, OASIS, ABIDE, and PPMI. Furthermore, the lack of comparison with other leading methods on these datasets makes it difficult to evaluate the model's effectiveness and its advancements over existing approaches. Additionally, the paper provides insufficient geometric explanation regarding the integration of Riemannian manifolds and symmetric positive definite (SPD) matrices, which may hinder understanding of the model's innovations. The rationale for choosing a geometric approach to analyze brain dynamics is also not clearly justified. Finally, the methodology for implementing the geometric components is inadequately discussed, which could impact reproducibility and practical application of the model.

**Questions:**

1. Insufficient Comparisons with State-of-the-Art Methods: The paper does not provide adequate comparisons with existing leading models on critical datasets such as ADNI, OASIS, ABIDE, and PPMI. This omission limits the ability to assess GeoMind's performance relative to established approaches and diminishes the impact of its contributions.

2. Lack of Detailed Geometric Explanation: The integration of Riemannian manifolds and symmetric positive definite (SPD) matrices within the GeoMind framework is not sufficiently explained. A clearer discussion of these geometric components and their relevance to the model would enhance understanding and strengthen the overall argument for its innovative approach.

3. Methodological Clarity and Reproducibility: The methodology for implementing the geometric aspects of the model is not adequately detailed, which could hinder reproducibility. Providing a more thorough description of the implementation and its effects on model performance would be beneficial for researchers looking to apply or build upon this work in practical settings.

---

> ### Author Response · Authors · 2024-11-21
> **Response to Reviewer QfXA**
>
> Thank you for acknowledging the contributions of our work. We are thrilled and grateful for your insightful feedback, which has significantly contributed to enhancing the quality of our manuscript. In the following responses, **W** and **Q** represent Weaknesses and Questions, and **A** represents the corresponding answer.
>
>
> **W \& Q1:** Insufficient Comparisons with State-of-the-Art Methods for ADNI, OASIS, ABIDE, and PPMI datasets.
>
>  **A1:** Thank you for your constructive comments, we have added several STOA methods to our comparison pool to provide adequate comparisons. First, we have added two dynamic-FC methods [1,2] on all datasets in the main manuscript (please refer to Table 2 in the updated manuscript), and we added three brain network analysis methods [3,4,5] on ADNI, OASIS, ABIDE and PPMI datasets in Appendix (please refer to Table 4).
>
> [1] STAGIN https://openreview.net/forum?id=X7GEA3KiJiH
>
> [2] NeuroGraph https://openreview.net/forum?id=MEa0cQeURw
>
> [3] BrainGNN https://www.sciencedirect.com/science/article/pii/S1361841521002784
>
> [4] BNT https://openreview.net/forum?id=1cJ1cbA6NLN
>
> [5] ContrastPool https://pubmed.ncbi.nlm.nih.gov/38656865/
>
> |                 | **ADNI**        | **OASIS**       | **PPMI**        | **ABIDE**       |
> |-----------------|-----------------|-----------------|-----------------|-----------------|
> | **BrainGNN**    | 76.57 ± 10.01   | 86.07 ± 5.71    | 67.88 ± 10.32   | 62.24 ± 4.44    |
> | **BNT**         | 79.68 ± 6.15    | 86.07 ± 3.19    | 64.55 ± 16.80   | 69.99 ± 5.37    |
> | **ContrastPool**| 80.08 ± 5.01    | 89.02 ± 4.22    | 69.78 ± 7.36    | 70.72 ± 3.45    |
> | **GeoMind**| **81.20 ± 2.27**    | **89.60 ± 1.87**    | **71.35 ± 10.26**    | **70.97 ± 3.60**    |
>
> It is clear that our GeoMind consistently outperforms the involved SOTA brain network analysis methods.
>
> **W \& Q2:** Lack of Detailed Geometric Explanation.
>
> **A2**: This comment is very helpful. Much appreciated. Since the human brain is a highly paralleled dynamic system, a system-level approach is in high demand to provide a holistic understanding of the mechanistic role of brain function in cognition. To address this challenge, we present our deep model that is designed to combine insight of mathematical principles (modeling the geometry of evolving functional connectivities on the manifold) and the power of machine learning (finding the best model from large-scale functional neuroimages). The conjecture is that we characterize the dynamics of the entire brain network as a whole by considering the $N \times N$ matrix as a data instance in the high-dimensional Riemanninal manifold of SPD matrices. Although the manifold algebra to operate the FC matrices is more complicated than conventional methods, we explore the new horizon of elucidating geometric relationships that might be essential for understanding complex brain dynamics. We have expanded the discussion (Appendix A.10) in the revised manuscript to detail how these geometric components contribute to both model accuracy and interpretability, strengthening the rationale for this approach.
>
> **W \& Q3:** Methodological Clarity and Reproducibility.
>
>  **A3:** Thank you for your constructive feedback. We have added more detailed implementation information in the Appendix (A.7 to A.9) to enhance clarity. Additionally, we would like to remind this reviewer that the full code has been made publicly available in the original manuscript, ensuring that other researchers can reproduce and build upon our work in practical applications https://anonymous.4open.science/r/GeoMind-12E8.

---

> > ### Author Response · Authors · 2024-12-02
> > **Kindly Reminder**
> >
> > Dear, Reviewer QfXA,
> >
> > We sincerely appreciate your time and effort in reviewing our manuscript and providing such constructive feedback. As the author-reviewer discussion phase is nearing its end, we wanted to check if you have any further comments or questions regarding our responses. We would be more than happy to continue the conversation if needed. If we have addressed your concerns, would you like to update your rating?
> >
> > Thank you so much.
> >
> > Best,
> >
> > Authors

---

### Official Review · Reviewer_pj61 · 2024-11-03

**Soundness:** 3
**Presentation:** 2
**Contribution:** 2
**Rating:** 5
**Confidence:** 4

**Summary:**

This paper presents GeoMind, a novel geometry deep model to learn spatial-temporal dynamics of brain functional networks, by the incorporation of state space model (SSM) and Riemannian manifold learning, considering the intrinsic geometric pattern of brain functional networks. Convolutional operation is adopted to reformulate the SSM to improve the inference efficiency. The proposed model has been evaluated using both brain connectome data and human action data.

**Strengths:**

The Riemannian manifold learning is introduced to the SSM to capture the spatial-temporal dynamics of brain functional networks.

**Weaknesses:**

1. The convolution operation on SPD. It seems that certain inductive biases of convolution on images do not hold for convolution on SPD, e.g., the locality of convolution assumes local image patterns are significant (local pixel patterns are similar). However, as the order of brain ROIs in brain atlas/connectome is kind of arbitrary, so that features in SPD at nearby locations may represent significantly different brain connectivity measures. This means the feature representation learned is difficult to interpret from the viewpoint of neuroscience. On the other hand, this means the feature representation learned may vary a lot if the order of brain ROIs is shuffled.

2. Lack of comparison with dynamic-FC based models. As the proposed method basically works on dynamic FCs, it should be compared with dynamic-FC based models, such as STAGIN (Kim et al., 2021), DynDepNet (Campbell et al., 2022), NeuroGraph (Said et al., 2023).

**Questions:**

1. The convolution operation on SPD. It seems that certain inductive biases of convolution on images do not hold for convolution on SPD, e.g., the locality of convolution assumes local image patterns are significant (local pixel patterns are similar). However, as the order of brain ROIs in brain atlas/connectome is kind of arbitrary, so that features in SPD at nearby locations may represent significantly different brain connectivity measures. This means the feature representation learned is difficult to interpret from the viewpoint of neuroscience. On the other hand, this means the feature representation learned may vary a lot if the order of brain ROIs is shuffled.
2. More implementation details are needed, such as kernel size in convolution and size of sliding window. It is also helpful to see how they will affect the performance.
3. In addition to feature/model interpretation, does the GaA module also improve the prediction?
4. As the proposed method basically works on dynamic FCs, it should be compared with dynamic-FC based models, such as STAGIN (Kim et al., 2021), DynDepNet (Campbell et al., 2022), NeuroGraph (Said et al., 2023).
5. In addition to manifold learning, it is also common to apply the tangent space projection to brain connectome data before feeding it into ML/DL models. Comparison with it would also be helpful to demonstrate the strength of learning on SPD.
6. In model comparison, BOLD signals were used as graph embeddings for the graph-based models. As BOLD signals are not directly comparable across different individuals (especially for resting-state fMRI), using static FC as embeddings may improve these models' performance.
7. In Fig.3, it looks like the region (orange circle) for HCP-WM is not in the dorsal attention region, please double check it.

---

> ### Author Response · Authors · 2024-11-21
> **Response to Reviewer pj61**
>
> ### We thank the reviewer's thorough feedback, which has greatly helped us improve the quality of our manuscript. We will address reviewer's concerns and questions one by one, where **W** and **Q** represents Weaknesses and Questions, and **A** represents the corresponding answer. ###
>
>  **W1 \& Q1:** ...the feature representation learned may vary a lot if the order of brain ROIs is shuffled...
>
> **A1:** We appreciate this insightful and constructive comment. **First of all**, we would like to highlight that the spatial pattern (such as edge and corner features) in a 2D image is completely different from the topological pattern (such as link and motifs) in a functional connectivity (FC) matrix. Since the data structures for image and graph differ fundamentally, the concept of image convolution with invariance to geometric distortions does not apply to the kernel convolution on FC matrices.
>
> **Second**, we appreciate the concerns raised by this reviewer regarding the possible issue of whether region shaffling might alter the learning outcomes. Below are our responses. From the perspective of network neuroscience, however, nearly all current studies adhere to pre-defined brain parcellations (also known as atlases). This approach ensures that findings can be compared across different studies. In this context, shuffling the regions during data analysis is not common. In addition, the feature representation learning and the attention component in our deep model are not sensitive to the actual shuffling of ROIs. To support this argument, we conducted an experiment on simulated data with ground truth. The results of the experiment are shown at https://anonymous.4open.science/r/GeoMind-12E8/simulated.pdf.
>
> Specifically, we use the SimTB toolbox (https://github.com/trendscenter/simtb )  to generate the simulated fMRI time series with three brain states (as shown in Figure left-middle, BOLD signals). As shown in Figure upper-left, the three brain states are characterized by functional connectivity (FC) matrices, each comprising three distinct modules (or communities) along the diagonal (state 1 in green, state 2 in blue, and state 3 in red). For each possible pair of nodes with the same module, the degree of connectivity is set to one, while no connections are permitted across modules, in alignment with previous studies (10.1109/TMI.2017.2780185 and 10.1016/j.neuroimage.2021.117791). This design reflects the small-world characteristics inherent to brain networks. We generate 2000 samples (including 1000 testing samples), where each sample has 300 time points and each brain state lasts 100 seconds.
>
> **After that**, to verify the stability of our GeoMind, we shuffle the brain regions, however, rather than using random shuffling methods, we rearrange the regions while maintaining a neuroscience-informed structural pattern. Specifically, the first six regions are swapped with the last four regions to create a new FC matrix (as shown in Figure lower-left).  After training our method, we visualize the identified network attention in Figure (right), it is clear that our tailored GaA module can capture the underlying pattern associated with the specific brain states, and these patterns are closely aligned with the change of network topology independent of the brain region ordering. We will include this result in the final version.
>
> Thank you for your constructive comment, your insight has been truly inspiring. While we initially employed convolution to accelerate SSM operations, we are more than willing to include this experiment in the manuscript if the reviewers deem it necessary. We sincerely appreciate your valuable comment.
>
> **W2 \& Q4:** Lack of comparison with dynamic-FC based models, such as STAGIN (Kim et al., 2021), DynDepNet (Campbell et al., 2022), NeuroGraph (Said et al., 2023).
>
> **A2:** Thank you for your constructive comment. We have added two dynamic-FC based models – STAGIN and NeuroGraph to our comparison pool (please refer to Table 2 in the updated revised manuscript). Note, since the DynDepNet is not public now (https://github.com/ajrcampbell/DynDepNet), we have contacted the authors to update the code,  when the code is public, we’d like to add it to our comparison pool. For a fair comparison, we use the default parameter settings for all methods and we report the 10-fold cross-validation results. The input configuration is consistent with our GeoMind approach, utilizing a window size of 15 and a step size of 1. The specific input is shown in line 1102 to line 1106 in the updated manuscript.

---

> ### Author Response · Authors · 2024-11-21
> **Response to Reviewer pj61**
>
> | Model             | Metric | HCP-WM        | ADNI         | OASIS         | PPMI            | ABIDE         |
> |--------------------|--------|------------------|-------------------|-------------------|----------------------|-------------------|
> | **STAGIN**        | Acc    |   91.05 ± 0.90   |   74.00 ± 5.13    |    88.97 ± 1.81   |     67.75 ± 8.65     |   69.36 ± 2.23    |
> |                   | Pre    |    91.11 ± 0.90  |    63.49 ± 15.47  |  89.33 ± 1.69    |    59.93 ± 13.32     |  69.94 ± 2.35    |
> |                   | F1     |   91.02 ± 0.90  |    65.50 ± 8.45   |    85.46 ± 2.89   |     60.22 ± 10.83    |    68.86 ± 2.35   |
> | **NeuroGraph**    | Acc    |     67.97 ± 1.41            | 77.60 ± 4.07      | 89.06 ± 2.05      | 73.31 ± 10.64    | 60.97 ± 2.00      |
> |                   | Pre    |        68.59 ± 1.17         | 76.19 ± 10.87     | _88.71 ± 3.15_    | 67.98 ± 14.56        | 62.98 ± 5.28      |
> |                   | F1     |     67.92 ± 1.33     | 73.52 ± 5.94      | 85.93 ± 2.32      | 68.63 ± 12.23        | 58.78 ± 4.26      |
>
> Our GeoMind achieves superior performance.
>
> **Q2:**  More implementation details are needed, such as kernel size in convolution and size of sliding window. It is also helpful to see how they will affect the performance.
>
> **A3:** Thank you for your constructive comment. The window size is set as 15, and step size is set as 1 in the original manuscript. We also conduct an ablation study on PPMI dataset in terms of window size.  We have made it clear in the updated revised manuscript (please refer to Table 6 in Appendix A.9).
>
> | **Window size** | **15**          | **25**          | **35**          | **45**          | **55**          |
> |------------------|-----------------|-----------------|-----------------|-----------------|-----------------|
> | **Acc**         | 71.35 ± 10.26   | 70.83 ± 15.74   | 71.69 ± 10.10   | 72.01 ± 8.51    | 71.01 ± 14.23   |
> | **Pre**         | 76.07 ± 7.33    | 74.72 ± 10.00   | 73.54 ± 6.50    | 71.73 ± 7.56    | 71.00 ± 7.89    |
> | **F1**          | 70.60 ± 9.73    | 71.71 ± 7.29    | 70.72 ± 3.90    | 68.56 ± 6.74    | 67.67 ± 7.18    |
>
> We can observe that GeoMind is relatively insensitive to window size, with optimal performance achieved at moderate window sizes, typically in the range of 25 to 35. This robustness can be attributed to GeoMind's reliance on the SSM module to capture dynamic temporal characteristics.
>
> **Q3:** In addition to feature/model interpretation, does the GaA module also improve the prediction?
>
> **A4:**  Thank you for your comment. We also conducted an ablation study about the w/o GaA module. Yes, the GaA module also improves the prediction performance.
>
> | **w/o GaA**     | **HCP-WM**      | **ADNI**        | **OASIS**       | **PPMI**        | **ABIDE**       |
> |------------------|-----------------|-----------------|-----------------|-----------------|-----------------|
> | **Acc**         | 97.25 ± 0.65 ↓   | 79.60 ± 2.80 ↓  | 89.26 ± 2.29 ↓  | 70.97 ± 8.02 ↓   | 69.75 ± 2.70 ↓   |
> | **Pre**         | 97.29 ± 0.64 ↓   | 80.51 ± 4.92  ↓ | 87.37 ± 5.68 ↓  | 73.53 ± 8.93  ↓  | 69.90 ± 1.68  ↓  |
> | **F1**          | 97.24 ± 0.66  ↓  | 76.86 ± 3.78  ↓  | 86.49 ± 3.52 ↓   | 67.34 ± 8.66 ↓   | 69.66 ± 1.24  ↓  |
>
> **Q5:** In addition to manifold learning, it is also common to apply the tangent space projection to brain connectome data before feeding it into ML/DL models.
>
> **A5:** Thank you for your comment, we agree with the reviewer’s thought, thus our comparison methods include the following situations：
>
> - For spatial models, such as GCN, we use the vector of brain connectome data (FC) as the input, the vectorization operation can be regarded as the tangent space projection.
> - For manifold-based models, such as SPDNet, we use the brain connectome data (FC) as the input, and then we apply tangent space projection on the learned feature representation to conduct downstream tasks (such as classification).
> - For sequential models, such as RNN, we use BOLD signals as the input.
> - For the dynamic-FC models, such as NeuroGraph, we employ a sliding window approach to generate a sequence of dynamic FC matrices, which are then used as inputs (graph).
>
> We have made it clear in the revised manuscript, please refer to Appendix A.7 and the experimental results are listed in Table 2 and Table 4.

---

> > ### Author Response · Authors · 2024-11-21
> > **Response to Reviewer pj61**
> >
> > **Q6:** Using static FC as embeddings may improve these models' performance.
> >
> > **A6:** Thank you for pointing this out, and we apologize for the confusion. We initially conducted experiments using both BOLD signals and vectorized static Functional Connectivity (FC) as embeddings for the graph-based models. Our findings, consistent with the results in NeuroGraph Table 2, indicate that using vectorized static FC as embeddings yields better performance than BOLD signals. Thus, the reported results are based on static FC embeddings. We have double-checked the code and results, which are consistent with the initial uploaded code line 55 - line 64,  https://anonymous.4open.science/r/GeoMind-12E8/ICLR2025/Comparsion_methods/Datasets_load.py).
> >
> > For sequential models, we use BOLD signals as inputs. We have clarified this distinction in the revised manuscript (Appendix A.7).
> >
> > Thank you again for helping us improve the clarity of our work.
> >
> > **Q7:** In Fig.3, it looks like the region (orange circle) for HCP-WM is not in the dorsal attention region, please double check it.
> >
> > **A7:**  Sorry for this confusion,  we have corrected it, the orange circle is in the central executive network (CEN), similar to the default mode network (DMN), these subnetworks are highly related to the working memory task (and have been frequently reported in neuroscience literature ( 10.1109/TMI.2022.3169640, 10.3389/fnhum.2020.00360, 10.1016/j.neuroimage.2013.05.079), please refer to Fig.3 in the revised manuscript for clarification. Thank you very much for your attention to this detail.

---

> ### Comment · Reviewer_pj61 · 2024-11-25
>
> Thank you for the responses and the additional results. While it is not common to shuffle brain regions in neuroimaging studies, the question is that the topological pattern of FC is not relevant to the order of brain regions, but the representations learned by convolutions on FC matrix are. The convolutions focus on a local receptive field, and they are dependent on the order of brain regions. As shown in the results on the simulated data, the changes in attention values are clear with a simple region rearrangement. A random shuffle may lead to larger changes. If the learned representations are not invariant to the order of brain regions, it may not capture the underlying topological pattern of FC well.

---

> ### Author Response · Authors · 2024-11-26
> **Eager to receive a response**
>
> Dear Reviewer,
>
> Thank you for your insightful comments. We deeply appreciate your perspective, as it has helped us reflect further on this important aspect of our work.
>
> We agree that convolutional operation might not be invariant to the spatial re-arrangement of input data (although our empirical experiments indicate our convolution operation is not sensitive to such spatial alteration). However, we DISAGREE that “*If the learned representations are not invariant to the order of brain regions, it may not capture the underlying topological pattern of FC well*”. Instead, we argue that even though we don’t have theoretical proof regarding the variance, our method does NOT have the issue of characterizing the relationship between topological patterns and cognitive tasks, for the following two reasons. (1) Since the brain atlas typically remains consistent throughout the study, the reviewer's concern holds less significance from the perspective of network neuroscience. (2) Invariance does not necessarily lead to performance drop.
>
> We noticed that the score has been reduced from ‘5’ to ‘3’. However, this reviewer stated, “The figures and explanations are clear and well-organized, making complex ideas easier to understand. The writing is also clear and easy to follow. The experiments are comprehensive, covering multiple datasets and a variety of comparison methods”. (**Is there any misunderstanding here?**) Since the unexpected drop of the score is not consistent with the overview of review comments, we would like to respectively ask for a justification.
>
> Over the past two weeks, we have worked diligently to address your concerns, as we greatly value your feedback and the opportunity to improve our work. We deeply respect the time and effort you have invested in reviewing our submission, and we kindly ask if you would give us another chance to address any remaining concerns or clarify any unresolved issues.
>
> Thank you so much.
>
> Best,
>
> Authors

---

> > ### Comment · Reviewer_pj61 · 2024-11-26
> >
> > I really appreciate the Authors' effort and hard work to improve the study. I adjusted the score due to concern about the convolution operation on the FC matrix. As mentioned in previous comments, the output of convolution for a certain brain region will be dependent on the order of brain regions (as its spatial locality will change). More validations are needed regarding this, especially on real data. Even there is no drop in performance, changes in attention values (or learned representations) may mislead the interpretation.
> >
> > This has also been discussed in previous study (BrainNetCNN: Convolutional neural networks for brain networks; towards predicting neurodevelopment, https://doi.org/10.1016/j.neuroimage.2016.09.046). I just copy the following sentence from the paper for reference, "However, spatial locality between entries of the adjacency matrix does not directly correspond to topological locality in the brain network. For an entry located at $A_{i,j}$, only those elements within the i-th row and j-th column are topologically local and so the typical grid convolutional filters used for images are not appropriate here".
> >
> > Please let me know if I have any misunderstanding here. Thanks.

---

> ### Author Response · Authors · 2024-11-26
> **Urgent Clarification, Kindly Reply**
>
> Dear, Reviewer pj61,
>
> Thank you for your insightful comment again. We have discussed your concern **in depth** and want to ensure we fully understand your question.
>
> Your concern seems to be: *“Should the attention remain unchanged when the region order is shuffled?”*  If our interpretation of your idea is not accurate, we would appreciate your clarification.
>
>
> First of all, we would like to clarify that **attention pattern (referring to the important task-specific brain regions)** is **INVARIANT** to the shuffling of region order.
>
> Shuffling the spatial positions of brain regions only modifies the **spatial distribution of these correlations** in the FC matrix. However, such spatial change does not affect the eigen-system (including eigenvalues as well as eigenvectors), In terms of manifold distance, the SPD matrix A and the counterpart matrix A’ after shuffling are identical, i.e., d(A, A’)=0. Since our attention mechanism and feature representation learning are grounded in the Riemannian manifold, the attention map used to **highlight the key brain regions** in classification tasks is expected to adjust its spatial position based on the pre-designed shuffling pattern. We have also validated this using the simulated data.
>
> In addition, the identified key brain regions **are ONLY relavent** to the **specific brain states** and are not influenced by the order of the brain regions. Consequently, the attention mechanism used to **highlight the importance of brain regions** in classification tasks **should** **effectively track** the truly significant regions corresponding to each brain state during the shuffling process.
>
> **Feature Invariance.** Simliar to attention maps that invariant with FC,  the high-dimensional semantic features learned by our model remians INVARIANT too. These high-level features exhibit spatial invariance concerning brain regions, meaning that alterations in the positions of brain regions do not affect the discriminative characteristics our model has captured. To validate the invariance of these learned features, we conducted classification on simulated data both before and after shuffling the brain regions. The results were remarkably similar, with classification accuracies of 97.33% (no shuffle) and 98.00% (after shuffle). These nearly identical outcomes empirically demonstrate that the discriminative features our model has learned remain unchanged despite the shuffling of brain region positions.
>
>
> Hope our clarification can address your concerns. If there is any misunderstanding, can you please give us a chance to clarify and improve?
>
>
> Thank you again for your thoughtful evaluation, and we hope to hear your perspective on how we might further refine our work.
>
> Best,
>
> Authors

---

> ### Author Response · Authors · 2024-11-26
> **Academic discussions with Reviewer pj61**
>
> Dear, Reviewer pj61,
>
> Thank you for your insightful response, **we enjoy academic discussions like this.**
>
> *“concern about the convolution operation on the FC matrix”*
>
> We would like to discuss the rationality (potential) for employing grid convolution in functional connectivity (FC) analysis from several perspectives.
>
> - **First**, grid convolution in natural image processing operates locally due to the inherent spatial relationships between adjacent pixels. Similarly, **this locality characteristic is present in FC data.** Real FC maps exhibit the **"small-world"** property regardless of the atlas used ( **brain networks have a "small-world" topology characterized by dense local clustering or cliquishness of connections between neighboring nodes** from Danielle S. Bassett [1]), resulting in **local block structures** that are highly discriminative (as shown in an example https://anonymous.4open.science/r/GeoMind-12E8/Example_FC.jpg). Consequently, applying grid convolution to FC effectively captures these local block features. There are a lot of works focusing on analyzing FC using traditional convolutional operations [2-4].
>
> [1]Small-world brain networks, https://journals.sagepub.com/doi/abs/10.1177/1073858406293182
>
> [2] https://ieeexplore.ieee.org/stamp/stamp.jsp?arnumber=8512372&casa_token=Kws_sofovQYAAAAA:Z2a1L9OOqzZ99aMJ9r3sKruHidEJAKKfww-BtOh-s5v3uqQeS0Vt-VN6aceDThPQ2IxiA3W_&tag=1
>
> [3] https://doi.org/10.1109/TMI.2024.3421360
>
> [4] https://doi.org/10.1016/j.brainresbull.2023.110846
>
> - **Second,** based on the aforementioned properties, it can be inferred that **completely random shuffling of brain regions would disrupt these local blocking structures, which do not exist in real data and contradict neuroscientific principles**. In our simulated dataset, we maintain the integrity of these blocks by only rearranging the order of entire blocks rather than randomly shuffling individual brain regions. This approach ensures that convolutional operations can continue to identify and extract meaningful local block structures.
>
> - **Finally,** it is important to highlight that **both our attention maps (which identify key brain regions) and the learned features remain invariant.** The task-relevant features are preserved despite block shuffling, and the significance of brain regions is maintained regardless of their spatial positions. As a result, **our attention maps consistently identify the most important brain regions for classification, irrespective of their placement within the FC.** (We kindly remind reviewer that we have provided a detailed explanation in our previous response.)
>
> - Again, we argue that the attention map pattern predicted by our model remains invariant to changes in region order. We acknowledge the reviewer's request for validation on real data and plan to conduct this experiment to substantiate our argument. Please note that the results of this experiment will NOT be included in the final version for two reasons: (1) such validation is out of scope, and (2) region shuffling is less likely the practical issue in neuroimaging studies.
> Hope our clarification can address your concern and we looking forward to your insightful perspective.
>
>
> Best,
>
> Authors

---

> > ### Comment · Reviewer_pj61 · 2024-11-26
> >
> > Thank you for the response and the relevant papers provided. However, it looks like that all the methods proposed in studies [2-4] adopt row-wise or column-wise convolution operations similar to that in BrainNetCNN, instead of using typical grid convolutions on local block. Please double check that.
> >
> > On the other hand, while the topology and properties (e.g., small-world) of FC are largely irrelevant to the atlas used, I think the local block structure is dependent on the atlas used.
> >
> > Thanks for your response again.

---

> ### Author Response · Authors · 2024-11-27
> **Can you please reconsider our work, thanks?**
>
> Dear Reviewer pj61,
>
> We sincerely appreciate your responsibility and patience. We are trying our best to address your concerns promptly.
>
> First, we’d like to provide more references about the convolutional operation on FC:
>
> [1] https://ieeexplore.ieee.org/stamp/stamp.jsp?arnumber=8512372&casa_token=Kws_sofovQYAAAAA:Z2a1L9OOqzZ99aMJ9r3sKruHidEJAKKfww-BtOh-s5v3uqQeS0Vt-VN6aceDThPQ2IxiA3W_&tag=1
>
> [2] https://doi.org/10.1016/j.brainresbull.2023.110846
>
> [3] https://doi.org/10.1016/j.media.2023.102921 (Fig. 2)
>
> Then, can we interpret the reviewer's comment as an acknowledgment that brain networks exhibit 'small world' properties? We are happy to provide additional references [4-8] to demonstrate that the 'small world' attribute is an inherent topological property of brain connectivity, independent of the choice of atlas. We acknowledge that while the global topological properties of FC, such as small-worldness, are generally consistent across different atlases, the local block structure may indeed exhibit variations depending on the atlas used. This is because the block structure reflects the grouping of regions and the spatial granularity defined by a particular atlas, thereby influencing the patterns observed within smaller subnetworks. In this context, our attention mechanism can capture the key brain regions relevant to the specific brain states, maintaining sensitivity to subtle variations in local structures, and convolution operations still hold potential for feature extraction based on blocking structures.
>
> [4] 10.1038/nrn2575
>
> [5] 10.1523/JNEUROSCI.3539-11.2011
>
> [6] 10.1016/j.neuroimage.2010.04.084
>
> [7] 10.1016/j.neuroimage.2009.10.003
>
> [8] 10.1523/JNEUROSCI.3539-11.2011
>
>
> If we address your last concerns, we kindly request that you re-consider our work, thank you so much.
>
> Have a nice day!
>
> Best,
>
> Authors

---

> > ### Comment · Reviewer_pj61 · 2024-11-27
> >
> > Thank you for the prompt response. The provided references are very informative.
> >
> > 1. For the convolutional operation on FC, this paper (ref [3], https://doi.org/10.1016/j.media.2023.102921) is quite interesting. It is mentioned in the paper (section 3.1) that "To make EdgeConv and NodeConv biologically meaningful, this study organizes the order of brain regions such that the brain regions in one functional subnetwork are next to each other. Fig. 1 illustrates the example of the brain functional network matrix with the order of brain regions ...". This suggests that the order of brain regions matters for convolution on FC. The default order of brain regions in some atlases follow the organization of functional subnetwork, but not every atlas.
> >
> > 2. The topology of FC (e.g., small-worldness) is invariant to the order of brain regions (even shuffled randomly for the same atlas), and I believe it will be largely consistent across different atlases. This has also been supported in ref [7] (10.1016/j.neuroimage.2009.10.003), "The order of nodes in connectivity matrices has no effect on computation of network measures but is important for network visualization (Fig.2A)". As shown in Fig.2A (ref [7]), the block structures in FC largely changed after "reorder nodes by modular structure", and the reordered FC demonstrated its topology more clearly than the counterpart with default order. I guess this kind of changes (or visualization) will affect the grid convolution on FC to a certain extent.
> >
> > Finally, I think the in-depth discussion is helpful to better understand the grid convolution on FC. While it would be great if grid convolution could effectively learn reproducible representations from FC (so that existing advanced CNN models could be used to explore FC straightforwardly), it is important to validate its robustness to certain factors (e.g., order of regions) in not only performance but also biological interpretation comprehensively. Thanks!

---

> ### Author Response · Authors · 2024-11-28
> **Can you please reconsider our work? Thanks.**
>
> Dear, Reviewer pj61,
>
> We sincerely appreciate your thoughtful feedback and the time you have dedicated to reviewing our work. We are doing our best to address your concerns promptly and thoroughly.
>
> We agree with that (“...The default order of brain regions in some atlases follow the organization of functional subnetwork, but not every atlas.”). However, to the best of our knowledge, brain parcellation is primarily guided by domain knowledge related to anatomical structures and functional organization. There is a converging consensus that anatomical regions in close proximity often exhibit functional connectivity patterns. Widely used brain atlases, such as AAL, Destrieux, Schaefer and so on, are not arbitrarily ordered. Instead, their organization systematically underlines the anatomical and functional principles.
>
> In extreme cases where brain regions are randomly shuffled, assume there is no discernible pattern, and there is no way to reorganize subnetworks. One solution in such scenarios is use a global convolution kernel with a size consistent with the total number of brain regions.
>
> Although such cases are exceedingly rare in practical applications, we are willing to follow the reviewer’s suggestion and validate our method on real human data if the reviewer thinks it necessary.
>
> We sincerely hope that our work can reach and inspire more researchers, as this could significantly enhance our understanding of brain networks. We also hope that you share our understanding of advancing research in this field.
>
> Thank you once again for your valuable comment.
>
> Sincerely,
>
> Authors

---

> > ### Comment · Reviewer_pj61 · 2024-11-28
> >
> > Thank you for the feedback.
> >
> > I agree with that it may have different effects on different atlas, and more validation or a principled way to reorder regions merit further investigation in the future. I think some discussion about this would be helpful in the paper.
> >
> > Thanks again for the in-depth discussion. I adjust the score back as my final rating. Hopefully the discussion will also be informative to all readers.

---

> > > ### Author Response · Authors · 2024-11-28
> > > **Thank you so much!**
> > >
> > > Dear Reviewer pj61,
> > >
> > > We sincerely appreciate your insightful comment and would be happy to include a discussion section in the final version. We greatly value and enjoy this constructive exchange.
> > >
> > > Thank you so much.
> > >
> > > Have a great day!
> > >
> > > Best,
> > >
> > > Authors

---

### Official Review · Reviewer_zeMB · 2024-11-04

**Soundness:** 3
**Presentation:** 3
**Contribution:** 2
**Rating:** 6
**Confidence:** 2

**Summary:**

This paper presents a novel geometric neural network, GeoMind, designed to capture brain dynamics by modeling functional connectivity on Riemannian manifolds. By leveraging state space models (SSM) with a focus on spatial and temporal brain dynamics, GeoMind aims to track evolving brain states effectively, addressing fMRI prediction problems. Additionally, the model's adaptability is validated through applications in human action recognition, showcasing its generalization capabilities.

**Strengths:**

1. The paper introduces an innovative method that combines manifold learning with SSM, effectively extending state space modeling beyond Euclidean spaces.
2. The figures and explanations are clear and well-organized, making complex ideas easier to understand. The writing is also clear and easy to follow.
3. The experiments are comprehensive, covering multiple datasets and a variety of comparison methods.

**Weaknesses:**

1. It seems that the paper does not clearly explain why it chose to use a Riemannian manifold for modeling brain dynamics. While this approach allows the model to capture certain geometric properties of brain data, it’s unclear why a simpler, more straightforward method wasn’t used instead. A stronger motivation, with an explanation of the specific benefits that the Riemannian manifold brings to neuroimaging tasks, would make the choice more convincing.
2. In my view, the main contribution of the paper lies in replacing the Euclidean algebra of conventional SSMs with Riemannian geometric algebra and applying this to fMRI images. However, the integration feels somewhat contrived, lacking a more natural or cohesive blend.

**Questions:**

1. Why was the Riemannian manifold chosen over simpler geometric spaces? Are there specific neuroimaging characteristics that necessitate this complex representation?
2. How does the choice of manifold-specific metrics like the Stein metric impact model performance compared to more straightforward metrics?

---

> ### Author Response · Authors · 2024-11-21
> **Response to Reviewer zeMB**
>
> ### Thank you for acknowledging the contributions of our work. We are thrilled and grateful for your insightful feedback, which has significantly contributed to enhancing the quality of our manuscript. In the following responses, **W** and **Q** represent Weaknesses and Questions, and **A** represents the corresponding answer. ###
>
> **W1:** ..Motivation..
>
> **A1:** Thank you for this great question. Human brain is a complex non-linear system. The dynamic nature of complex system cannot be understood by thinking of the system as comprised of independent elements. Rather, an approach is needed to utilize knowledge about the complex interactions within a system to understand the behavior of the system overall. In light of this, modeling the fluctuation of functional connectivities on the Riemannian manifold provides a holistic view of understanding how brain function emerges in cognition and behavior.  We have further clarified these motivations in the revised manuscript (please refer to line 107 to line 110).
>
> **W2:**  In my view, the main contribution of the paper lies in replacing the Euclidean algebra of conventional SSMs with Riemannian geometric algebra and applying this to fMRI images. However, the integration feels somewhat contrived, lacking a more natural or cohesive blend.
>
> **A2:**  We partially agree with this comment. One of our major contributions is the integration of SSM into the manifold learning scenario, which mainly lies in the realm of methodology innovation. Prior to the method development, we are motivated to explore a novel research framework to understand the complex relationship between brain function and cognition/behavior that combines the power of machine learning and insight of mathematical principles. In our opinion, the well-defined biological underpinning is another significant contribution. To that end, we presented a manifold-based deep model of SSM that allows us to establish a holistic understanding of functional fluctuations, where the characteristic evolution of FC matrices is described by the latent spate-state model in the Riemannian manifold. Our experiments demonstrate that our manifold-based deep model yields improved performance and interpretability in capturing complex brain dynamics. We hope this clarifies the rationale behind our approach, and we have revised the manuscript to better convey the reasoning and coherence of this integration.
>
> **Q1:** Why was the Riemannian manifold chosen over simpler geometric spaces? Are there specific neuroimaging characteristics that necessitate this complex representation?
>
> **A3:** Thank you for this excellent question. Functional connectivity (FC) is commonly used to model brain dynamics. Traditional methods often represent FC as a vector by flattening the matrix. However, this approach disrupts the intrinsic geometric structure of the brain’s connectivity patterns, which is essential to preserve for accurate representation. Since the FC matrix is derived from correlations between brain regions, it forms a symmetric positive definite (SPD) matrix with well-defined mathematical properties. Meanwhile, the Riemannian manifold of SPD matrix is a well-studied area with many successful applications in computer vision and biomedicine fields. Taken together, we present the deep model ($GeoMind$) to capture remarkable geometric patterns in functional neuroimages, enabling more meaningful analysis that aligns with the natural structure of brain connectivity data.
>
> **Q2:** How does the choice of manifold-specific metrics like the Stein metric impact model performance compared to more straightforward metrics?
>
> **A4:** Thank you for this insightful question. We have not specifically evaluated the impact of manifold-specific metrics since the driving force of our method is to maximize the correlation between connectome feature representations (learned on the manifold) and the outcome labels (such as cognitive states and diagnosis labels). In this regard, the manifold metric is NOT used in our deep model. Instead, we project the feature representations to the tangent space and then plug them into a fully connected network, where we use conventional cross-entropy as the loss function.

---

> > ### Comment · Reviewer_zeMB · 2024-11-26
> >
> > Thank you for your response and the clarification. I appreciate the additional insights provided and would like to maintain my score.

---

> > > ### Author Response · Authors · 2024-11-26
> > > **Response to Reviewer zeMB**
> > >
> > > Dear Reviewer zeMB,
> > >
> > > We really appreciate that.
> > >
> > > Have a great day!
> > >
> > > Best,
> > >
> > > Authors

---

### Official Review · Reviewer_FUYW · 2024-11-05

**Soundness:** 3
**Presentation:** 2
**Contribution:** 3
**Rating:** 6
**Confidence:** 2

**Summary:**

The paper introduces GeoMind, a geometric neural network model that uses state space models to track brain dynamics on a Riemannian manifold. By modeling brain connectivity with symmetric positive definite matrices, it preserves the geometric structure of brain activity. A geometric-adaptive attention mechanism helps identify critical brain connections relevant to diseases like Alzheimer's and Parkinson’s.

**Strengths:**

GeoMind outperforms most competing models in both brain activity and action recognition tasks. This high performance shows its effectiveness and robustness across different applications.

**Weaknesses:**

1. Although the model includes an attention mechanism, the manifold-based learning approach and complex neural architecture could still make it challenging to interpret results fully, especially for clinical use.

2. The use of Riemannian geometry and manifold learning may introduce additional computational overhead, which could be a drawback in large-scale clinical settings.

**Questions:**

1. How does the manifold-based approach improve insights into brain behavior? Does it reveal connections between brain structure and specific cognitive outcomes that simpler models miss?

2. fMRI data can be noisy due to patient movement or equipment issues. Has GeoMind been tested with noisy or incomplete data? Are there any built-in ways to handle this?

---

> ### Author Response · Authors · 2024-11-21
> **Respone to Reviewer FUYW**
>
> ### Thank you for acknowledging the contributions of our work. We are thrilled and grateful for your insightful feedback, which has significantly contributed to enhancing the quality of our manuscript. In the following responses, **W** and **Q** represent Weaknesses and Questions, and **A** represents the corresponding answer. ###
>
> **W1:** Although the model includes an attention mechanism, the manifold-based learning approach and complex neural architecture could still make it challenging to interpret results fully, especially for clinical use.
>
> **A1:** Thank you for your insightful comment. We fully understand the importance of model explainability in clinical applications.  To that end, we introduce a geometric attention mechanism (GaA) to provide interpretability in terms of node-wise association with clinical outcomes.
>
> Meanwhile, the formulation of the entire brain network as a manifold data instance offers another system level of understanding of how brain functional dynamics contribute to the biological mechanism of disease progression. Specifically, our manifold-based deep model of SSM is designed to capture complex data structures (such as dynamic functional connectivity) that are otherwise difficult to model in Euclidean space, providing a mathematical insightful representation of temporal characteristics relevant to clinical outcomes. We have also conducted an ablation study to verify the contribution of the attention module (Table 6 bottom in the updated manuscript shows the numerical results).
>
> | **w/o GaA**     | **HCP-WM**      | **ADNI**        | **OASIS**       | **PPMI**        | **ABIDE**       |
> |------------------|-----------------|-----------------|-----------------|-----------------|-----------------|
> | **Acc**         | 97.25 ± 0.65 ↓   | 79.60 ± 2.80 ↓  | 89.26 ± 2.29 ↓  | 70.97 ± 8.02 ↓   | 69.75 ± 2.70 ↓   |
> | **Pre**         | 97.29 ± 0.64 ↓   | 80.51 ± 4.92  ↓ | 87.37 ± 5.68 ↓  | 73.53 ± 8.93  ↓  | 69.90 ± 1.68  ↓  |
> | **F1**          | 97.24 ± 0.66  ↓  | 76.86 ± 3.78  ↓  | 86.49 ± 3.52 ↓   | 67.34 ± 8.66 ↓   | 69.66 ± 1.24  ↓  |
>
> **We believe that**, combined with interpretability techniques, our approach can still offer valuable information while maintaining a degree of interpretability suitable for clinical contexts.
>
> **W2:** The use of Riemannian geometry and manifold learning may introduce additional computational overhead, which could be a drawback in large-scale clinical settings.
>
> **A2:** Thank you for your comment. We agree that Riemannian geometry might introduce additional computational overhead due to the computationally intensive Singular Value Decomposition (SVD) involved. However, the running time (2.51ms/item) in the application stage is well suited to clinical settings (please see the analysis of inference time and model parameters in Appendix A.8). Note, we listed the inference time of large-scale dataset HCP-WM (17,296).
>
> **Q1:** How does the manifold-based approach improve insights into brain behavior? Does it reveal connections between brain structure and specific cognitive outcomes that simpler models miss?
>
> **A3:** We appreciate these great questions. We expect our manifold-based deep model to facilitate our understanding on brain behavior in the following ways.
>
> - (1) Enhance the prediction accuracy. A plethora of neuroscience findings indicate that fluctuation of functional connectivities exhibits self-organized spatial-temporal patterns. Following this notion, we conceptualize that well-defined mathematical modeling of intrinsic data geometry of evolving functional connectivity (FC) matrices might be the gateway to enhance prediction accuracy.  Our experiments have shown that respecting the intrinsic data geometry in method development leads to significantly higher prediction accuracy for cognitive states, as demonstrated in Table 2.
>
> - (2) Enhance the model explainability. We train the deep model to parameterize the transition of FC matrices on the Riemannian manifold (Eq. 4 &5). By doing so, we are able to analyze the temporal behaviors with respect to each cognitive state using post-hoc complex system approaches such as dynamic mode decomposition, stability analysis. Due to the page limit, we haven’t included these results in the current version. However, we have added this to the discussion section in Appendix A.10.
>
> - (3) Provide a high-order geometric attention mechanism that is beyond node-wise or link-wise focal patterns. Conventional methods often employ attention components for each region or link in the brain network separately, thus lacking the high-order attention maps associated with neural circuits (i.e., a set of links representing a sub-network). In contrast, the geometric attention mechanism (Eq. 6) in our method operates on the Riemannian manifold, taking the entire brain network into account. As shown in Fig. 3, our method has identified not only links but also sub-networks relevant to cognitive states and disease outcomes.

---

> ### Author Response · Authors · 2024-11-21
> **Respone to Reviewer FUYW**
>
> **Q2:** fMRI data can be noisy due to patient movement or equipment issues. Has GeoMind been tested with noisy or incomplete data? Are there any built-in ways to handle this?
>
> **A4:** Thank you for your insightful question. Yes, fMRI is low Signal-to-Noise Ratio (SNR) data, as summarized in SSM to neuroimaging application (Appendix A.1), SSMs naturally incorporate probabilistic structures, allowing them to effectively handle noisy or uncertain data. This is particularly advantageous in low SNR datasets, such as fMRI (https://ieeexplore.ieee.org/stamp/stamp.jsp?arnumber=8715509) and Electroencephalogram (EEG) data (https://ieeexplore.ieee.org/document/8609948), where the ability to account for noise and uncertainty is critical. Plus, the equivalent convolution operation serves as a filter, helping to mitigate this issue. To validate our approach, we have conducted emotion recognition experiments on three widely used EEG datasets: SEED, DEAP, and HCI. Based on the previous results, we report the results of the top three methods as follows:
>
> | **Mamba**     | **SEED**      | **DEAP**        | **HCI**       |
> |------------------|-----------------|-----------------|-----------------|
> | **Acc**         | 81.62 ± 0.48    | 88.98 ± 1.79    | 86.89 ± 4.56    |
> | **Pre**         | 79.76 ± 0.95    | 88.10 ± 1.83    | 75.51 ± 17.15    |
> | **F1**          | 76.31 ± 0.45    | 87.79 ± 2.07    | 69.78 ± 14.23    |
> | **SPDNet**     |    |     |    |
> | **Acc**         | 85.32 ± 1.41    | 90.23 ± 2.40    | 93.16 ± 2.73    |
> | **Pre**         | 82.68 ± 1.88    | 88.69 ± 3.64    | 84.62 ± 6.21    |
> | **F1**          | 80.16 ± 1.92    | 88.48 ± 3.28    | 76.78 ± 2.54    |
> | **GeoMind**     |       |         |        |
> | **Acc**         | 92.51 ± 2.39    | 93.92 ± 1.46    | 95.23 ± 3.59    |
> | **Pre**         | 92.62 ± 2.36    | 93.94 ± 1.46    | 95.05 ± 3.74    |
> | **F1**          | 92.48 ± 2.39    | 93.91 ± 1.46    | 94.92 ± 4.05    |
>
> It is clear that our method can also achieve promising results.

---

> > ### Author Response · Authors · 2024-12-02
> > **Kindly Reminder**
> >
> > Dear, Reviewer FUYW,
> >
> > We sincerely appreciate your time and effort in reviewing our manuscript and providing such constructive feedback. As the author-reviewer discussion phase is nearing its end, we wanted to check if you have any further comments or questions regarding our responses. We would be more than happy to continue the conversation if needed. If we have addressed your concerns, would you like to update your rating?
> >
> > Thank you so much.
> >
> > Best,
> >
> > Authors

---

### Author Response · Authors · 2024-11-21
**General response to all reviewers**

We sincerely thank all reviewers for their thoughtful and constructive feedback. We have addressed all concerns raised by the reviewers (**highlighted in gray in the updated manuscript**) and kindly request that the reviewers consider our rebuttals.

Also, we appreciate the recognition of our contributions, practical value, experimental rigor and presentation. Notably, the reviewers highlighted:

**(1) Technical Novelty**

- The paper introduces an **innovative method** that combines manifold learning with SSM, effectively extending state space modeling beyond Euclidean spaces. (**Reviewer zeMB**)
- This **innovative approach** provides a unique framework for modeling complex neural activities, with promising results in identifying specific brain states from task-based fMRI data. (**Reviewer QfXA**)

**(2) Comprehensive Experiments**

- GeoMind outperforms **most competing models** in both brain activity and action recognition tasks. This **high performance** shows its **effectiveness and robustness** across different applications. (**Reviewer FUYM**)
- The experiments are **comprehensive**, covering multiple datasets and a variety of comparison methods. (**Reviewer zeMB**)
- This **innovative approach** provides a unique framework for modeling complex neural activities, with **promising results** in identifying specific brain states from task-based fMRI data. (**Reviewer QfXA**)


**(3) Practical Value**

- Additionally, the model shows **potential for early diagnosis** of neurodegenerative diseases such as Alzheimer’s, Parkinson’s, and Autism, which underscores its relevance in **clinical applications**. (**Reviewer QfXA**)
- This high performance shows its effectiveness and robustness across **different applications**. (**Reviewer FUYM**)

**(4) Presentation**
- The figures and explanations are **clear and well-organized**, making complex ideas easier to understand. The writing is also clear and easy to follow. (**Reviewer zeMB**)

Thank you so much for your time and consideration.

---

> ### Author Response · Authors · 2024-11-27
>
> Dear AC and Reviewers,
>
> We hope that our rebuttal has addressed your concerns and now we provide more clarity and understanding of our work in light of your valuable feedback.
>
> We sincerely appreciate your thoughtful comments and invite you to reach out with any additional questions or requests for clarification.
>
> In addition, we had an in-depth discussion with Reviewer pj61 regarding an open question that influenced the reviewer’s revised rating. We have provided thorough explanations and responses to address the reviewer’s concerns and sincerely hope that Reviewer pj61 will reconsider the evaluation. We remain committed to resolving all concerns and also **kindly invite the other reviewers and the AC to join this discussion.**
>
> We hope to contribute to the esteemed ICLR and the broader ML/Neuroscience community by having this paper published.
>
> Thank you for your time and consideration.
>
> Best,
>
> Authors

---

### Note · Authors · 2025-01-22

I have read and agree with the venue's withdrawal policy on behalf of myself and my co-authors.